# Process Simulation of High-Pressure Nanofiltration (HPNF) for Membrane Brine Concentration (MBC): A Pilot-Scale Case Study

**DOI:** 10.3390/membranes15040113

**Published:** 2025-04-04

**Authors:** Abdallatif Satti Abdalrhman, Sangho Lee, Seungwon Ihm, Eslam S. B. Alwaznani, Christopher M. Fellows, Sheng Li

**Affiliations:** 1Water Technologies Innovation Institute and Research Advancement (WTIIRA), Saudi Water Authority (SWA), Al-Jubail 31951, Saudi Arabia; swihm78@daum.net (S.I.); ealwaznani@swcc.gov.sa (E.S.B.A.); cfellows@une.edu.au (C.M.F.); sli@swcc.gov.sa (S.L.); 2School of Civil and Environmental Engineering, Kookimin University, 77 Jeongneung-ro, Seongbuk-gu, Seoul 02707, Republic of Korea; 3School of Science and Technology, University of New England, Armidale 2351, Australia

**Keywords:** membrane brine concentration (MBC), high-pressure nanofiltration (HPNF), mathematical model, process simulation, response surface methodology (RSM)

## Abstract

The growing demand for sustainable water management solutions has prompted the development of membrane brine concentration (MBC) technologies, particularly in the context of desalination and minimum liquid discharge (MLD) applications. This study presents a simple model of high-pressure nanofiltration (HPNF) for MBC. The model integrates reverse osmosis (RO) transport equations with mass balance equations, thereby enabling acceptable predictions of water flux and total dissolved solids (TDS) concentration. Considering the limitations of the pilot plant data, the model showed reasonable accuracy in predicting flux and TDS, with R^2^ values above 0.99. The simulation results demonstrated that an increase in feed flow rate improves flux but raises specific energy consumption (SEC) and reduces recovery. In contrast, an increase in feed pressure results in an increased recovery and brine concentration. Increasing feed TDS decreases flux, recovery, and final brine TDS and increases SEC. Response surface methodology (RSM) was employed to optimize process performance across multiple criteria, optimizing flux, SEC, recovery, and final brine concentration. The optimal feed flow rate and pressure vary depending on the criteria in the improvement scenarios, underscoring the importance of systematic process improvement.

## 1. Introduction

The global water crisis has created an urgent need for the implementation of sustainable water management solutions. Industrialization and urbanization are significantly increasing pressure on global water resources [1]. The uneven availability of water, in conjunction with the effects of climate change, presents a considerable challenge in guaranteeing a sustainable supply of freshwater [2]. This has increased reliance on desalination technologies to address water demands in regions with scarce resources.

Despite this potential of desalination, there are considerable concerns regarding the environmental impact of brine discharge and the carbon footprint of desalination processes [3,4,5,6,7,8,9]. These two issues are interconnected: To reduce the environmental impact of concentrates, it is necessary to reduce the amount of discharged brine and to recover valuable resources from it [10,11,12,13,14]. However, these processes of brine treatment and mining require considerable energy inputs, which in turn result in an increase in carbon emissions [11,15,16]. Due to these challenges, concerns have emerged regarding the long-term feasibility of desalination [6,17,18,19].

One potential solution to this challenge is the membrane brine concentration (MBC), which refers to a membrane-based process for the further concentration of brine or saline water [20,21]. In contrast to thermal processes such as multi-effect distillation (MED) and mechanical vapor compression (MVC), MBC does not require a phase change, thereby reducing energy consumption [22]. Furthermore, MBC achieves a higher water recovery ratio than that of conventional seawater reverse osmosis (SWRO) and has the potential to be used for desalination brine mining, enabling the recovery of valuable resources [23]. For example, a novel hybrid system comprising nanofiltration (NF), seawater reverse osmosis (SWRO), and MBC has recently been proposed as a means of reducing the volume of brine and recovering salts such as NaCl and magnesium [21,24].

High-pressure nanofiltration (HPNF), also known as low-salt rejection reverse osmosis (LSRRO), represents a novel approach to the implementation of MBC [25,26,27]. This methodology employs “loose” RO or NF membranes to treat highly saline water at elevated hydraulic pressures. These membranes with a lower salt retention than conventional RO membranes can be used to reduce the required hydraulic pressure (transmembrane pressure) for the concentration process [28]. In contrast to conventional reverse osmosis (RO) systems, HPNF is configured to concentrate brine in a more efficient manner, particularly in contexts such as minimal and zero liquid discharge (MLD/ZLD). This technology enables higher water recovery ratios and improved energy efficiency, rendering it a more practical and cost-effective solution for the treatment of hypersaline brines [26]. For example, HPNF can achieve specific energy consumption (SEC) as low as 2.4 to 8.0 kWh/m^3^, significantly lower than traditional thermal methods such as mechanical vapor compression (MVC), which consumes 20–25 kWh/m^3^ [25]. In addition, there are many other advantages of MBC over MVC [21]. This process is particularly useful for economically reducing the volume of brine, thereby recovering more fresh water and leaving behind a more concentrated salt solution for potential recovery [29]. For example, the final NaCl concentration by RO-electrodialysis (ED) was higher (244 g/L) than that of the NF-SWRO-HPNF (225), but its energy consumption was 30~44% higher [24,30]. Other ED systems reported a lower NaCl concentration with a higher energy consumption [31]. The scalability of HPNF for large-scale industrial applications was discussed in our previous study [24].

In the development of HPNF (LSRRO), mathematical modeling and process simulation have become indispensable tools for optimizing operational conditions and membrane configurations [32]. The models for HPNF are used to predict system performance, including flux, permeate quality, and energy efficiency [26]. To elucidate the operating principles of HPNF, a theoretical model based on mass balance was developed and applied to two-, three-, and four-stage HPNF configurations [25]. Membrane transport equations based on a solution–diffusion model were combined with the mass balance equations to elaborate the effect of membrane properties and operating conditions on the performance of a multi-stage HPNF [27]. A method to determine this maximum allowable NaCl retention was developed to show that current RO and NF membranes are not suitable for HPNF [28]. A cost optimization model for HPNF was also developed by combining several sub-models such as the membrane stage model, NaCl solution property model, and cost model [33]. The performance of LSRRO was theoretically compared with other process configurations such as split-feed counterflow reverse osmosis (SF-CFRO), cascading osmotically mediated reverse osmosis (COMRO), and osmotically assisted reverse osmosis (OARO) [26,34]. The majority of these studies have focused on theoretical analysis, either without experimental data or with only limited data from laboratory-scale tests [25]. Since few models were developed based on data collected from real HPNF plant data [28,35], it is difficult to use previous models for practical implementation.

The aim of this study is to develop a mathematical model and conduct simulations for an HPNF process for MBC. The model was established by combining the RO transport models with process mass balance equations. The model results were compared with operational data from a pilot-scale HPNF system and applied for wide application such as process simulation and improvement. Subsequently, a series of HPNF simulations were conducted under various conditions to provide valuable insights into the key factors influencing the process efficiency, including membrane characteristics, operating conditions, and feed water composition. Unlike previous models, our model was developed based on a real pilot plant for HPNF. Moreover, the model contains only essential equations and is therefore simple, requiring a minimum number of parameters determined by experiments. In addition, the model performs process improvement based on response surface methodology (RSM). To the best of the authors’ knowledge, no previous studies have conducted a theoretical analysis based on pilot plant data for the HPNF process for MBC.

Our study aims to contribute to this broader objective by focusing on the MBC process using HPNF. While comprehensive improvement of the entire desalination plant, including energy consumption (SEC) and effluent valorization, is crucial, this study is positioned as a focused step within this larger framework. The rationale for isolating the MBC process in our study is twofold: (1) to provide a theoretical evaluation of HPNF performance in brine concentration, which remains a key challenge in enhancing overall desalination efficiency, and (2) to establish a foundation for integrating MBC into broader system-level improvement in future studies.

## 2. Model Development

### 2.1. Process Configuration

The process configuration, which consists of NF, SWRO, and HPNF, was selected for model development. Our previous study analyzed the applicability and effectiveness of this system for brine concentration [24].

Figure 1 illustrates a process flow diagram for this system. The diagram begins with a feed stream directed into a nanofiltration (NF) unit, which separates multi-valent ions from seawater. After this initial filtration step, the permeate is supplied to a SWRO unit. The SWRO unit produces further separation, yielding a second permeate stream, and a more concentrated brine. The brine from the SWRO stage is then directed into a series of HPNF stages, labeled as 1st HPNF, 2nd HPNF, and 3rd HPNF. These stages work sequentially to further concentrate the brine while extracting permeates at each stage. The areas of the membranes in the HPNF stages are denoted as S_m,1_, S_m,2_, and S_m,3_. The final product of the process is a highly concentrated brine from the 3rd HPNF stage, with permeates collected from multiple stages, which can be reused or disposed of depending on the situation. The permeate from the 1st HPNF returns to the SWRO stage and is mixed with the NF permeate. More information and justification of this process configuration are available in our previous study [24].

### 2.2. Membrane Transport Equations for HPNF

A model for the HPNF stages, which are shown inside the dashed box in Figure 1, was developed by combining the membrane transport models and the mass balance equations. The NF and SWRO stages are not included in this model, as numerous existing models already address these processes. The general equation describing water transport and pressure drop in these processes is [35,36,37,38,39](1)$$J_{w}=A\left(\frac{P_{f1}+P_{f2}}{2}-P_{p}-\left(\frac{\pi_{f1}+\pi_{f2}}{2}-\pi_{p}\right)\right)=A\left(\Delta P-\Delta \pi \right)$$(2)$$J_{s}=B\left(c_{m}-c_{p}\right)=J_{w}c_{p}$$(3)$$\phi =\frac{\left(c_{m}-c_{p}\right)}{\left(c_{f}-c_{p}\right)}=e^{\frac{J_{w}}{k_{m}}}$$(4)Sh=0.065Re0.875Sc0.25(5)Re=ρvdhμ(6)$$\mathrm{Sh}=\frac{k_{m}d_{h}}{D}$$(7)$$\mathrm{Sc}=\frac{\mu }{\rho D}$$(8)$$d_{h}=\frac{4\varepsilon }{\frac{2}{h}+\left(1-\varepsilon \right)S_{V,SP}}$$(9)$$\lambda =\frac{2P_{drop}}{\rho v^{2}}\frac{d_{h}}{l_{m}}$$(10)λ=6.23Re−0.3(11)$$P_{f1,i+1}=P_{f1,i}-P_{drop}$$
where *J_w_* is the water flux (L/m^2^-h), *A* is the water permeability of the membrane (L/m^2^-h-bar), *P_f1_* is the hydraulic pressure on the feed inlet, *P_f_*_2_ is the hydraulic pressure on the feed outlet, *P_p_* is the hydraulic pressure on the permeate side, Δ*P* is the transmembrane pressure between the feed and permeate, *π_f_*_1_ is the osmotic pressure on the feed inlet, *π_f2_* is the osmotic pressure on the feed outlet, *π_p_* is the osmotic pressure on the permeate side, *Δπ* is the transmembrane osmotic pressure between the feed and permeate, *B* is the salt permeability of the membrane (L/m^2^-h), *c_m_* is the concentration of the solute on the membrane surface, *c_p_* is the concentration of the solute in the permeate, *c_f_* is the concentration of the solute in the feed, *ϕ* is the concentration polarization modulus, *k_m_* is the mass transfer coefficient, Sh is the Sherwood number, Re is the Reynolds number, Sc is the Schmidt number, ρ is the density (kg/m^3^), v is the crossflow velocity (m/s), dh is the hydraulic diameter (m), μ is the viscosity (Pa-s), *D* is the solute diffusion coefficient (m^2^/s), ε is the porosity of the spacer, λ is the friction coefficient, *S_V,SP_* is the specific surface of the spacer, *h* is the channel height (m), *P_drop_* is the pressure drop (Pa), *l_m_* is the module length (m), and *P_f_*_1,*i*_ is the feed pressure at the *ith* module.

The model is focused on brine concentration without considering other contaminants, such as organic compounds. Although it may reduce its versatility, it could not be included because of a lack of data on the concentrations of such contaminants in the pilot plant. In addition, the feed water of MBC processes is pretreated by NF, which can remove most organic compounds [21,24]. In our model, the water and salt permeabilities (*A* and *B*) are assumed to be constant with time. This means that the effect of membrane lifetime is not included. Although this effect is important, it could not be considered because there is no information on the lifetime of HPNF membranes due to the lack of long-term operation studies. Accordingly, it should be noted that our current model is applicable to the initial state of HPNF operation.

### 2.3. Mass Balance in HPNF

To simulate a pilot-scale HPNF system, it is necessary to consider mass balance equations for water and the salt. In fact, it comprises several stages, and the mass balance equations should be solved in each stage [40,41].(12)$$Q_{f,i}=Q_{b,i}+Q_{p,i}$$(13)$$c_{f,i}Q_{f,i}=c_{b,i}Q_{b,i}+c_{p,i}Q_{p,i}$$
where *Q_f,i_* is the flow rate of the feed supplied to the ith stage (m^3^/hr), *Q_b,i_* is the flow rate of the brine from the ith stage (m^3^/hr), *Q_p,i_* is the flow rate of the permeate produced from the ith stage (m^3^/hr), *c_f,i_* is the concentration of the feed supplied to the ith stage (g/L), c*_p,i_* is the concentration of the brine from the ith stage (g/L), and c*_p,i_* is the concentration of the permeate produced from the ith stage (g/L).

As shown in Figure 1, the feed to the HPNF stage consists of two streams: the brine from the previous stage and the permeate from the next stage. Accordingly, the flow rate and the concentration of the feed are given as(14)$$Q_{f,i}=Q_{b,i-1}+Q_{p,i+1}=Q_{b,i-1}+J_{w,i+1}S_{m,i+1}\;\;\;(i=1,\;2)$$
(15)$$c_{f,i}=\frac{c_{b,i-1}Q_{b,i-1}+c_{p,i+1}Q_{p,i+1}}{Q_{f,i}}=\frac{c_{b,i-1}Q_{b,i-1}+J_{s,i+1}S_{m,i+1}}{Q_{f,i}}\;\;\;(i=1,\;2)$$
(16)$$Q_{p,i}=J_{w,i}S_{m,i}$$(17)$$c_{p,i}Q_{p,i}=J_{s,i}S_{m,i}$$
where *Q_b,i−1_* is the flow rate of the brine from the (*i* − 1)th stage (m^3^/hr), *Q_p,i+_*_1_ is the flow rate of the permeate produced from the (*i* + 1)th stage (m^3^/hr), c*_b,i−_*_1_ is the flow rate of the brine from the (*i* − 1)th stage (g/L), c*_p,i+_*_1_ is the concentration of the permeate produced from the (*i* + 1)th stage (g/L), *J_w,i_* is the water flux in the ith stage, *J_w,i_* is the salt flux in the ith stage, and *S_m,i_* is the membrane area of the *ith* stage. In the 3rd stage, *Q_f,_*_3_ = *Q_b,_*_2_, and *c_f,_*_3_ = *c_b,_*_2_.

The average water flux is calculated by(18)$$J_{avg}=\frac{\sum_{i=1}^{3}Q_{p,i}}{\sum_{i=1}^{3}S_{m,i}}=\frac{\sum_{i=1}^{3}J_{w,1}S_{m,i}}{\sum_{i=1}^{3}S_{m,i}}$$

The total energy required for the HPNF stages includes the energy necessary for pressurizing the feed of the 1st HPNF, the permeate of the 2nd HPNF, and the permeate of the 3rd HPNF. This results in the following equations:(19)$$E_{total}=Q_{f,1}P_{f,1}+Q_{p,2}P_{f,1}+Q_{p,3}P_{f,2}$$

Here, the specific energy consumption (SEC) is defined as the energy requirement to produce the permeate supplied to the SWRO stage. Accordingly, SEC is calculated as(20)$$SEC=\frac{E_{total}}{Q_{p,1}}=\frac{Q_{f,1}P_{f,1}+Q_{p,2}P_{f,1}+Q_{p,3}P_{f,2}}{Q_{p,1}}$$

Considering the system boundary in Figure 1, the water recovery (*RR*) of the HPNF system corresponds to the ratio of the permeate for the 1st HPNF (*Q_p_*_,1_) to the feed for the 1st HPNF (*Q_b_*_,0_):(21)$$RR=\frac{Q_{p,1}}{Q_{b,0}}=\frac{J_{w,1}S_{m,1}}{Q_{b,0}}$$

As shown in Figure 1, the final brine concentration (c*_b,HPNF_*) is the same as the c_b,3_. The above equations were solved using the EES software V11.698 (F-Chart, Madison, WI, USA). A desktop PC (13th Gen Intel^®^ I7-13700K, 128 GB RAM) was used to run the program.

## 3. Materials and Methods

### 3.1. Pilot Plant and Operating Conditions

A series of pilot-scale studies were conducted at the pilot plant facility in Jubail, Saudi Arabia. The 3-stage HPNF system employs 4-inch OsmoARO^TM^ membranes from FTS H2O (HBR-TFC-4040, Fluid Technology Solutions, Albany, OR, USA). According to the manufacturer, the maximum operating pressure is 1150 psi (80 bar), and the maximum temperature is 113 °F (45 °C). Other information on membrane durability is not available. The specifications of the HPNF membranes are summarized in Table 1.

As shown in Figure 1, the HPNF pilot plant consists of three stages. The numbers of the elements in the 1st, 2nd, and 3rd HPNF stages were 24, 16, and 8, respectively. Each vessel contains eight elements. Accordingly, the numbers of vessels in the 1st, 2nd, and 3rd HPNF stages are 3, 2, and 1, respectively. The vessels in each stage were arranged in parallel. The applied pressure at the inlet of the 1st stage was 70 bar. Each stage operated in constant pressure mode, which was carried out by regulating the pressure of the HPNF feed pump. The pressure drops along the 4 modules were less than 0.5 bar. The booster pump pressures were set to be the same as the feed pump pressure. The process configuration of the HPNF stages is illustrated in Figure 1. More information on the pilot plant, including the operating conditions and data reproducibility, is available in our previous study [24].

### 3.2. Feed Water Characteristics

Table 2 shows a comparison of the main water quality parameters for seawater and RO brine, the latter used as feed for the high-pressure nanofiltration (HPNF) system. As the seawater has been treated through the NF and SWRO stages, the RO brine shows increased concentrations of major ions such as sodium (Na⁺), chloride (Cl^−^), and total dissolved solids (TDS). For example, TDS increases from 45.0 g/L in seawater to 76.29 g/L in the RO brine, and NaCl concentration increases from 39.2 g/L to 73.89 g/L. In addition, the relative concentration of NaCl (NaCl/TDS) is higher in the RO brine (97%) than in the seawater (87%), reflecting the efficiency of the NF stage in removing divalent ions from the seawater prior to the RO stage.

### 3.3. Analytical Methods

The major ions and properties were analyzed in the SWA-WTIIRA laboratory under international test methods, specifically ASTM D1293 for pH, ASTM 2540C for TDS (gravimetric method), and ASTM 3120B. For sulfate, the spectrophotometer was employed, while for bicarbonate, titration was utilized in accordance with AWWA 2310. AWWA 4110 A was used for chloride, and for the remaining ions, ICP-OES was employed.

## 4. Results and Discussion

### 4.1. Operation of Pilot Plant

As the first step, the operating data were obtained from the HPNF pilot plant. Figure 2a depicts the relationship between flow rate and operational time (hours) for a range of streams within the system, including feed, permeate, and brine. The flow rates across different streams range from approximately 0.46 m^3^/h to 4.78 m^3^/h over a period of 2000 h. The flow rate of the feed to the HPNF stages, which corresponds to *Q_b_*_,0_, exhibits lower flow between 2.80 and 3.14 m^3^/hr. As the number of HPNF stages increases, the permeate flow rate decreases from 2.63 m^3^/hr for the 1st HPNF stage to 0.58 m^3^/hr for the 3rd HPNF stage. The final brine flow rate showed an average of 0.48 m^3^/hr. This results in an overall HPNF stage recovery of 89%.

Figure 2b illustrates the total dissolved solids (TDS) concentration over time for various streams in the HPNF stages. The feed to the HPNF stages exhibits the TDS between 7.76 × 10^4^ mg/L and 8.32 × 10^4^ mg/L. The TDS in the permeate of the 1st HPNF stage (*C_p_*_,1_) ranges between 4.58 × 10^4^ mg/L and 5.48 × 10^4^ mg/L. The TDS in the permeate of the 2nd HPNF stage (*C_p_*_,2_) remains around 1.07 × 10^5^ mg/L, and that of the 3rd HPNF stage (*C_p_*_,3_) stays around 1.54× 10^5^ mg/L. In the end, the TDS in the final brine (*C_b_*_,3_) increases up to 2.19× 10^5^ mg/L. This corresponds to a 2.7-fold increase in the TDS from the initial feed (*C_b_*_,0_) to the final brine (*C_b_*_,3_).

### 4.2. Comparison of Model Calculation with Pilot Plant Data

The result of model calculation was compared with the pilot plant data for the HPNF stages. The water and salt permeabilities of the membrane, which were obtained based on the preliminary results in single-element test equipment, were 3.148 × 10^−12^ m^2^-s/kg and 9.074 × 10^−6^ m/s, respectively. Figure 3a shows the relationship between the predicted flow rate and the actual flow rate measured in the pilot plant. The data points are distributed closely along the diagonal line, which represents a good agreement between the predicted and observed values. This demonstrates an acceptable correlation between the model and experimental results. The high *R*^2^ value for the linear regression (0.991) suggests that the developed model reasonably predicts flow rates, suggesting the model’s usefulness in simulating the performance for the HPNF process.

Figure 3b compares the model predictions for total dissolved solids (TDS) concentrations (in mg/L) with the actual TDS values measured in the pilot plant. Similar to the flow rate comparison, the data points closely follow the diagonal line, indicating strong agreement between model predictions and pilot plant data. The model prediction is less accurate at higher TDS values. This is probably because the measurement of the TDS in the pilot plant was less accurate than that of the flow rate. The high TDS values were measured in the 3rd HPNF stage, which has more uncertainties than the other stages, leading to larger fluctuations (see Figure 2b). Overall, the prediction of TDS is reasonably accurate (*R*^2^ = 0.995), confirming the capability of the model to simulate the HPNF process. Accordingly, the model may be used as a reasonable tool for optimizing HPNF operations and designing efficient brine concentration systems.

It should be noted that the model results were compared to the pilot plant at a constant feed pressure of 70 bar. Due to restrictions imposed by the manufacturer of the pilot plant, it was not possible to freely adjust the feed pressure. Therefore, it is insufficient to validate the model using these data. Further work will be required to fully verify the model with pilot plant data under a wider range of operating conditions.

### 4.3. Simulation of Pilot Plant Performance

To understand the effect of operating variables on the performance of the HPNF process, a set of simulations was conducted using the developed model. Three operating variables, including the feed flow rate, feed TDS, and feed pressure, were selected for the simulation. Before the simulation, it was confirmed that the feed and permeate flow rates used in calculations correspond to existing/realistic 4-inch spiral modules/vessels. All the conditions for the simulation were selected based on the conditions of the real pilot plant. The one-factor-at-a-time method (OFAT), including the assessment of variables, or factors, one at a time instead of multiple factors simultaneously, was used. The standard condition for OFAT is set to the feed flow rate of 3.0 m^3^/hr, feed TDS of 80,000 mg/L, and feed pressure of 70 bar. In each test of one variable, the other variables are fixed at the standard condition. The performance of the HPNF process was evaluated in terms of the average flux, SEC, overall recovery, and final brine concentration. The pump efficiency was assumed to be 0.7 for the SEC calculation. The pump efficiency in the model was set to be similar to that in the pilot plant [24]. The efficiency is relatively low because the pump capacity was about 100~125 m^3^/day. The efficiencies of such small pumps are much lower than those of large pumps used in large-scale desalination plants (~0.85).

Figure 4 illustrates the results of the process simulation to investigate the effect of feed flow rate on the performance of the HPNF process. Figure 4a shows the dependency of the average flux and specific energy consumption on the feed flow rate. Both the average flux and SEC increase linearly as the feed flow rate increases from approximately 2.5 to 4.0 m^3^/h. The average flux rises from around 11 L/m^2^·h to nearly 13.5 L/m^2^·h, indicating that higher flow rates enhance the water permeation through the membrane. Similarly, SEC values increase from about 5.6 to 6.8 kWh/m^3^, reflecting the higher energy demand associated with increased flow rate. While an increase in the feed flow rate enhances the flux, it simultaneously leads to an escalation in energy consumption. This indicates that a balance between flux and energy consumption is essential for the optimal design of the process.

Figure 4b examines the impact of feed flow rate on the overall recovery and final brine total dissolved solids. The recovery rate decreases with an increase in the feed flow rate, with a range of approximately 0.85 to 0.70. The final brine TDS decreases with higher flow rates, from approximately 2.5 × 10^5^ mg/L to 1.7 × 10^5^ mg/L. The salt removal, which is defined as the relative amount of salt removed from the system in the brine from the total amount fed to the HPNF cascade, decreases from 0.551 to 0.312 with an increase in the feed flow rate. These results indicate that higher flow rates dilute the final brine, reducing brine concentration, recovery, and salt removal. Therefore, these results emphasize the need to carefully optimize flow rates to achieve a balance between maximizing recovery and minimizing energy consumption while maintaining high brine concentration in the HPNF process.

The relationship between feed TDS, average flux, and specific energy consumption is illustrated in Figure 5a. As the feed TDS increases from 70,000 mg/L to 90,000 mg/L, the average flux decreases linearly from approximately 11.9 L/m^2^·h to 11.1 L/m^2^·h. This reduction in flux is attributed to the higher osmotic pressure caused by increased feed TDS, which reduces water flux through the membrane. SEC also increases with rising feed TDS, ranging from 5.6 to 6.4 kWh/m^3^, reflecting the fact that the higher energy demand is associated with processing more concentrated feed water. Figure 4b shows the effect of feed TDS on recovery and final brine TDS. As the feed TDS increases, recovery decreases from approximately 0.83 to 0.73, indicating reduced water recovery efficiency with higher salinity feeds. This trend is consistent with the behavior of flux, where increased feed TDS limits water transport through the membrane. Meanwhile, the final brine TDS decreases with an increase in feed TDS, changing from around 2.05 × 10^5^ mg/L to 1.96 × 10^5^ mg/L. The salt removal shows a similar trend to the final brine TDS, decreasing from 0.489 to 0.404. This can be attributed to the fact that the negative effects of reduced flow and subsequent recovery are more significant than the positive effects of an increased feed concentration. These findings suggest that feed TDS has a negative impact on the performances of the HPNF, including flux, SEC, recovery, final brine concentration, and salt removal.

Figure 6 presents the results of a process simulation investigating the effect of feed pressure on the performance indices for the HPNF process. As shown in Figure 6a, both flux and SEC increase with an increase in the feed pressure. The flux rises from 0.8 L/m^2^·h to nearly 12.1 L/m^2^·h, and SEC increases from 5.8 to 6.2 kWh/m^3^ as the feed pressure increases from 63 bar to 77 bar. The increasing trend in both flux and SEC clearly indicates that more water can be produced with higher energy consumption by applying a higher driving force. As illustrated in Figure 6b, there is a positive correlation between the feed pressure and both recovery and final brine concentration. As feed pressure increases, the recovery increases from 0.71 to 0.83, and the final brine TDS increases from 1.66 × 10^5^ mg/L to 2.38 × 10^5^ mg/L. At the same time, the salt removal is improved from 0.394 to 0.492. These results indicate that increasing the feed pressure is an effective method for enhancing the recovery rate, brine concentration, and salt removal.

According to these results, the SEC of HPNF ranges from 5.8 kWh/m^3^ to 6.8 kWh/m^3^, which is significantly lower than those of thermal desalination techniques applicable for brine concentration. For example, the total carbon footprints of multi-stage flash (MSF), multi-effect distillation (MED), and SWRO were reported to be 18.0, 14.7, and 4.7 kg CO_2_-eq, respectively. Assuming that the carbon emission of membrane processes is almost proportional to their energy consumption and that HPNF may have about 2~2.5 times higher SEC than conventional SWRO, the carbon footprint of HPNF is about 9~14 kg CO_2_-eq. Nevertheless, it is necessary to conduct more rigorous analysis of the carbon footprint of HPNF in comparison with other competing techniques.

### 4.4. Application of Response Surface Methodology

Although the OFAT approach is beneficial for understanding the effect of operating variables on process performance, it is inadequate for analyzing the interactions among factors and responses and is time-consuming to find the optimum conditions [42]. Consequently, a more comprehensive approach based on response surface methodology (RSM) was employed to address the shortcomings of the OFAT approach. There are several advantages of RSM over OFAT [43]: First, RSM allows for the simultaneous variation of multiple factors, enabling the study of interactions between these factors. Second, RSM can identify optimal conditions using fewer experimental runs compared to OFAT. In addition, RSM employs statistically designed experiments, which include replication, randomization, and blocking.

Table 3 presents the design matrix and simulation results for RSM applied to the HPNF system. The ranges of values in the RSM were determined based on the operating conditions of the pilot plant. The feed flow rate should be set between 2.6 and 3.3 m^3^/hr due to the capacity of the pump in the pilot plant. The TDS of the brine was given to be nearly constant at approximately 80,000 mg/L. The feed pressure was set to be constant in the pilot, but it was assumed to be adjustable between 68 and 72 bar. Wider ranges were impractical for the improvement of the operating conditions of the pilot plant.

The table lists 20 experimental runs with three key operational factors: feed flow rate, feed TDS, and feed pressure. The corresponding responses include average flux, SEC, recovery, and final brine TDS. This design matrix is based on a central composite design (CCD) approach, a widely used RSM technique that allows the development of quadratic regression models to understand the interactions and quadratic effects of the operational parameters on the system’s performance. The use of CCD provides a balanced distribution of points that ensures the efficient estimation of both linear and higher-order effects, facilitating the improvement of process conditions. In the RSM analysis, the three factors (feed flow rate, feed TDS, and feed pressure) were varied across different levels to explore their influence on the four system responses. Commercial software (Design Expert, Minneapolis, MN, USA) was used to create the table, and a series of simulation runs using the model were conducted based on the conditions in this table.

Table 4 presents the summary of fit statistics and the regression equations for the RSM models developed in this study. The table provides key statistical metrics, including the model F-value, predicted R^2^, adequate precision, and the regression equations for each response variable: flux, specific energy consumption (SEC), recovery, and final brine TDS. The F-values for all responses are highly significant, with flux having an F-value of 38,824, SEC at 341,700, recovery at 81,057, and final brine TDS at 19,611.7, indicating that the developed models are statistically robust and reliable for predicting system performance. The predicted R^2^ values for all responses are close to 1, ranging from 0.9996 to 1.0000, which demonstrates excellent model accuracy and the ability to predict responses with high precision. The adequate precision values, ranging from 477.78 to 2227.56, are all well above the threshold value of 4, confirming that the models have a strong signal-to-noise ratio and are suitable for navigating the design space. The regression equations provided in the table quantify the relationship between the operational factors and the responses. Each response is expressed as a function of linear, interaction, and quadratic terms, capturing the complex interactions between factors and their non-linear effects on system performance. These regression equations can serve as predictive models that can be used to optimize the HPNF process by identifying the factor combinations that lead to the best performance in terms of maximizing flux and recovery while minimizing energy consumption and final brine TDS.

Figure 7 presents a set of response surface plots illustrating the combined effects of feed flow rate and feed pressure on four key performance indices of the HPNF process: average flux, SEC, recovery, and final brine concentration. The operating variables, such as the feed flow rate and feed pressure, were varied while the feed TDS was fixed at 80,000 mg/L. This is because the feed TDS was assumed to be a fact that cannot be adjusted within the boundary of the HPNF process.

As shown in Figure 7a, the flux increases as both feed flow rate and feed pressure rise, with higher pressures driving greater flux through the membrane. The highest flux values (around 12.5 L/m^2^·h) occur at the upper range of both flow rate (around 3.3 m^3^/h) and pressure (around 75 bar). This trend suggests that higher operating pressures and flow rates improve water permeation through the membrane. In Figure 7b, SEC increases with both higher flow rates and pressures, with the most significant increases occurring in the upper range of both variables. This demonstrates the energy cost trade-off when optimizing for flux, as higher fluxes achieved by increasing feed flow rate and pressure come at the expense of greater energy consumption. Figure 7c,d show the effects on recovery and final brine concentration, respectively. Recovery increases with feed pressure but decreases slightly with higher feed flow rates, indicating that lower flow rates are desired to increase the recovery. The final brine concentration decreases as flow rate increases but increases slightly with the feed pressure. While these plots provide insight into the relationship between the operating variables and performance indices, they highlight the need to balance flux, energy consumption, recovery, and brine concentration for optimal system operation.

The dependence of the four variables on the feed flow rate and the feed pressure are summarized in Figure 8. With an increase in the feed flow rate, the average flux and SEC increase, but the recovery and final brine concentration decrease. With an increase in the applied feed pressure, the values for all four variables increase. This implies that there are trade-off relationships between these variables. Accordingly, a multi-criteria improvement is required to meet the target conditions for the four variables at the time.

### 4.5. Multi-Criteria Improvement

Using the regression model in Table 3, the operating conditions for the HPNF process can be optimized to satisfy multiple objectives. To demonstrate the multi-criteria improvement, a set of scenarios were prepared as summarized in Table 5. The average flux (Javg), SEC, and recovery (RR) are defined in Equations (18), (20), and (21). The final brine concentration (c*_b,HPNF_*) is identical to c_b,3_ as shown in Figure 1. Each scenario has specific target conditions for average flux, specific energy consumption (SEC), recovery, and final brine concentration. In each scenario, an ideal balance between different performance indices was explored using the regression model. For example, in Scenario 1, the goal is to achieve an average flux of at least 11 L/m^2^-h while keeping the SEC below 6.5 kWh/m^3^, with a minimum recovery of 0.75 and a final brine concentration of at least 200,000 mg/L. To increase the flux, it is necessary to increase both the feed flow rate and pressure. However, this also increases the SEC, which fails the criteria. Furthermore, the recovery and final brine concentration also reduce with an increase in the feed flow rate, which may also fail their criteria.

Figure 9 illustrates the results of multi-criteria improvement of feed flow rate and feed pressure for the HPNF process, based on process simulations. The yellow regions in each plot represent the combinations of feed flow rate (m^3^/h) and feed pressure (bar) that satisfy the performance criteria for the four scenarios outlined in Table 4. The overlay plots are generated using the Design-Expert software, which integrates the responses of average flux, specific energy consumption (SEC), recovery, and final brine concentration to identify the optimal operating conditions for each scenario. These yellow regions in the subfigures highlight the ranges of feed flow rate and pressure where the system meets the specified targets for the respective scenarios. In Figure 9a, the yellow area is located predominantly at high feed pressures and low flow rates. On the other hand, the yellow region in Figure 9b shifts to lower feed pressures and lower flow rates. These lower-pressure conditions help reduce energy consumption (<6 kWh/m^3^) while still meeting the recovery target (>0.8). The yellow area in Figure 9c is more restricted to higher feed pressures and flow rates, which is needed to achieve the high flux target (>12 L/m^2^-h). The feed pressure should be higher than 72 bar, and the feed flow rate should be higher than 3.3 m^3^/hr in this scenario. In Figure 9d, the yellow area is located at the upper extremes of both feed flow rate and pressure, with feed pressures higher than 73 bar and flow rates higher than 2.95 m^3^/h. These conditions of high pressure and low flow rate allow the system to achieve a high level of brine concentration (>230,000 mg/L). In summary, the variation in the location and size of the yellow areas in the subfigures reflects the trade-offs between different performance metrics and emphasizes the importance of selecting operating conditions for the specific goals of the HPNF process.

## 5. Conclusions

In this study, the HPNF process for MBC was theoretically investigated to understand and optimize its operation. The following conclusions were drawn:A mathematical model for HPNF was developed for MBC and compared with pilot plant data. Considering the limitations of the pilot plant data, the model showed reasonable accuracy in predicting flux and TDS, with R^2^ values above 0.99.Simulations showed that the lower feed flow rates and higher feed pressures favored recovery and brine concentration. Increasing flow rate and feed pressure improved flux at the expense of energy efficiency. An increase in feed TDS reduces flux, recovery, and final brine TDS and increases SEC.Response surface methodology (RSM) was used to optimize flux, SEC, recovery, and brine concentration. The F-values for all responses range from 81,057 to 341,700. The predicted R^2^ values for all responses are close to 1, and the adequate precision values are all well above the threshold value of 4. These results indicate that the developed models are statistically reliable.Using the RSM models, the optimum conditions were explored for four typical scenarios of MBC operation. Depending on the criteria in the scenarios, the optimal feed flow rate and pressures vary. For instance, the feed flow rate should be low, and pressure should be intermediate when high recovery (≥0.80) and low SEC (≤6.0 kWh/m^3^) are required (scenario 2). On the other hand, the feed flow rate and pressure should be high to achieve high flux (≥12 L/m^2^-h) (scenario 3). This suggests the importance of systematic process improvement for MBC.

Since this study focuses on the HPNF stage, the interconnected nature of the entire desalination process (NF-SWRO-HPNF) is not considered. The limitations of this study can be summarized as follows:Instead of considering the recovery of the whole process, only the recovery of the HPNF was considered in this work. The first NF and SWRO stages were treated as boundary conditions. Accordingly, the current mode is not intended to optimize the entire NF-SWRO-HPNF system. Future work will explicitly integrate variable NF/RO performance into a holistic optimization framework.The SEC of the overall process was not included in the model. This study prioritizes the HPNF stage because the energy consumption of HPNF is much higher than that of SWRO and NF. Nevertheless, future work is needed to optimize the SEC of the whole system by a holistic approach.The purpose of this study was to develop a practical, industrially relevant tool for brine concentration, balancing computational efficiency with predictive utility. Future work will be needed to incorporate spatially resolved parameters and dynamic effects.

The current manuscript does not include a cost-of-ownership analysis or economic feasibility evaluation of HPNF MBC. This is because it is beyond the scope of this work. Techno-economic analysis results for the NF-RO-MBC were obtained in our previous work [12], although a different MBC process was considered. Further work will be required to conduct an economic analysis of the HPNF MBC.

## Figures and Tables

**Figure 1 membranes-15-00113-f001:**
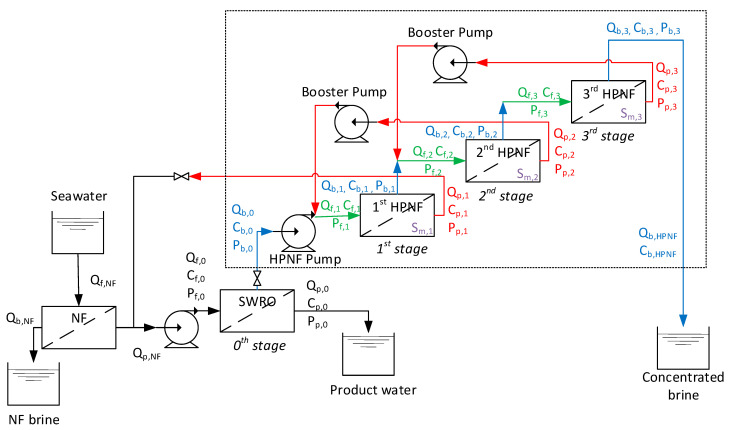
Schematic diagram of the high-pressure nanofiltration (HPNF) process for membrane brine concentration (MBC). The symbols for feed, brine, and permeate are marked in green, blue, and red, respectively.

**Figure 2 membranes-15-00113-f002:**
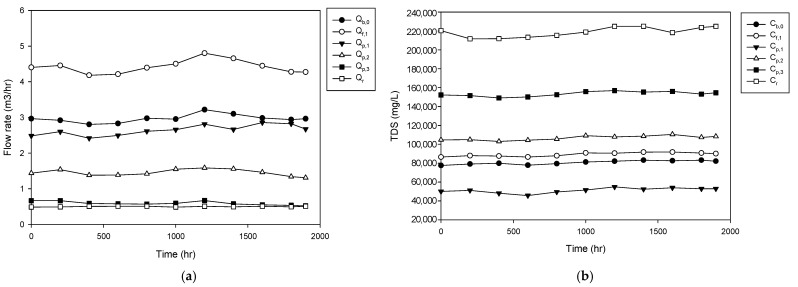
Pilot plant operation of the high-pressure nanofiltration (HPNF) process for membrane brine concentration (MBC). (**a**) Flow rate. (**b**) TDS.

**Figure 3 membranes-15-00113-f003:**
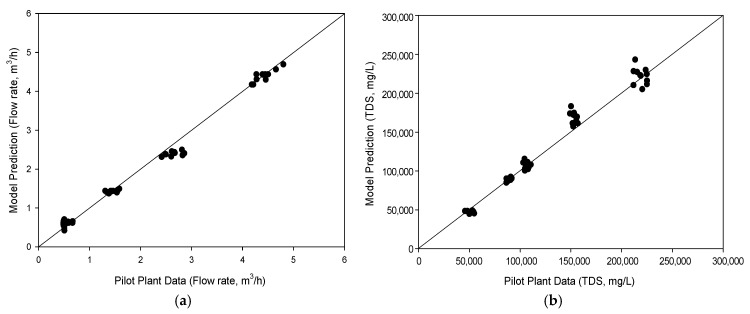
Comparison between model predictions and pilot plant data for the high-pressure nanofiltration (HPNF) process. (**a**) Flow rate. (**b**) TDS.

**Figure 4 membranes-15-00113-f004:**
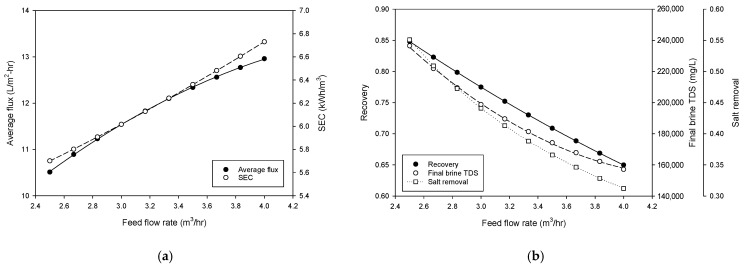
Process simulation for the effect of feed flow rate on (**a**) average flux and SEC and (**b**) recovery and final brine concentration.

**Figure 5 membranes-15-00113-f005:**
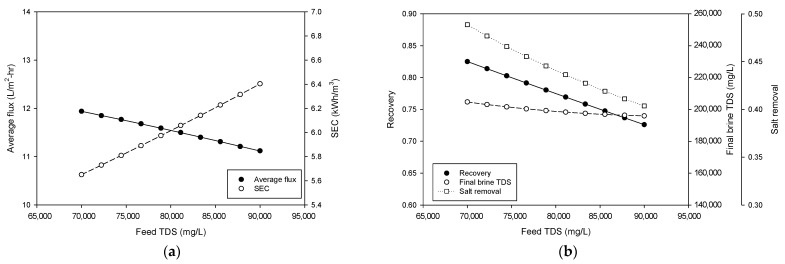
Process simulation for the effect of feed TDS on (**a**) average flux and SEC and (**b**) recovery and final brine concentration.

**Figure 6 membranes-15-00113-f006:**
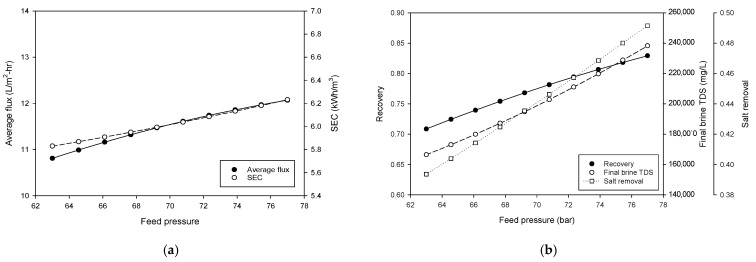
Process simulation for the effect of feed pressure on (**a**) average flux and SEC and (**b**) recovery and final brine concentration.

**Figure 7 membranes-15-00113-f007:**
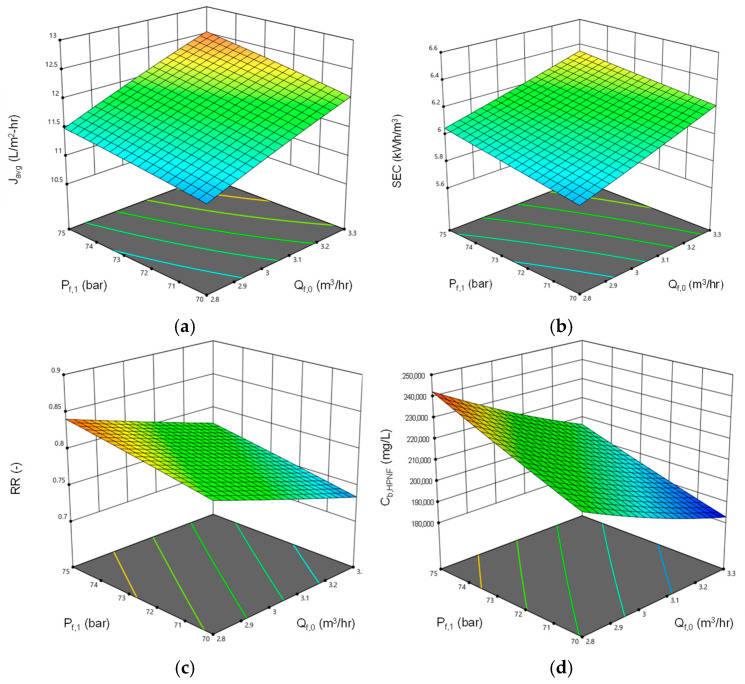
Process simulation for the combined effect of feed flow rate and feed pressure on (**a**) average flux (*J_avg_*), (**b**) SEC, (**c**) recovery (*RR*), and (**d**) final brine concentration (*C_b,HPNF_*).

**Figure 8 membranes-15-00113-f008:**
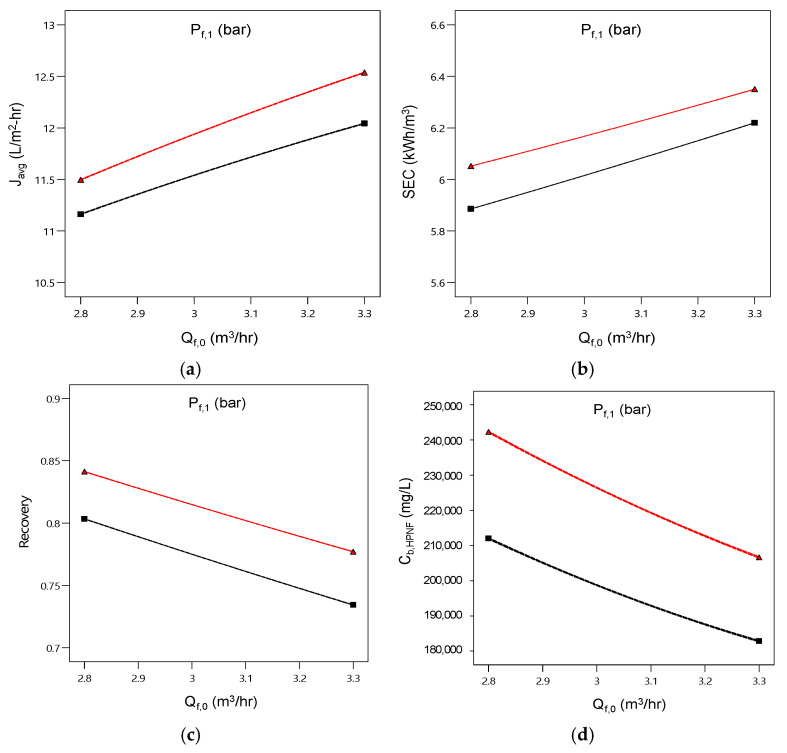
Process simulation for the combined effect of feed flow rate and feed pressure on (**a**) average flux (*J_avg_*), (**b**) SEC, (**c**) recovery (*RR*), and (**d**) final brine concentration (*C_b,HPNF_*). The black and red curves correspond to 70 bar and 75 bar, respectively.

**Figure 9 membranes-15-00113-f009:**
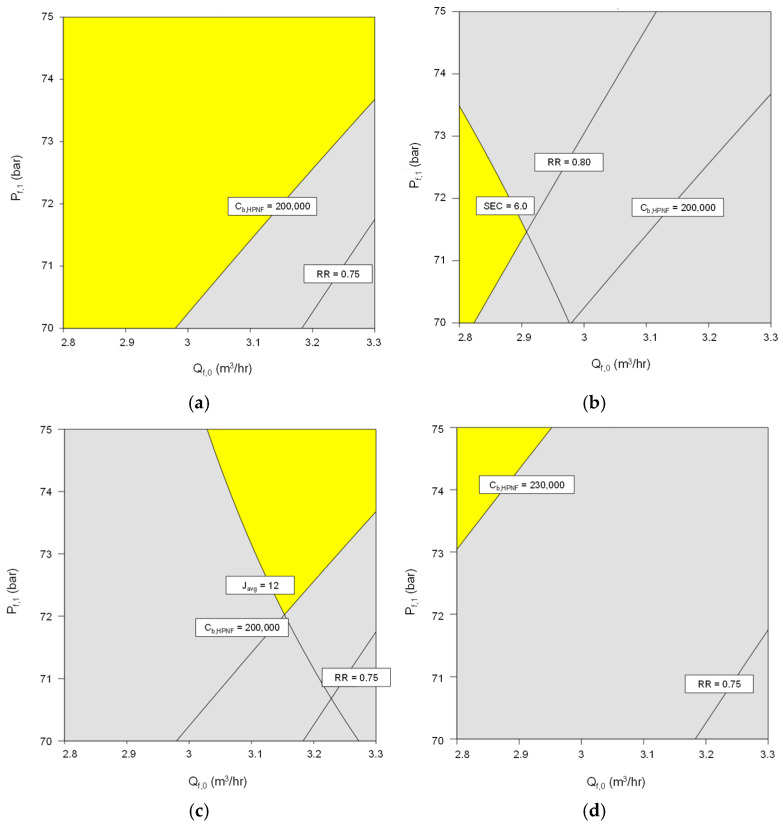
Improvement of feed flow rate and pressure based on process simulation. The yellow areas correspond to the operation conditions to meet (**a**) scenario 1, (**b**) scenario 2, (**c**) scenario 3, and (**d**) scenario 4, respectively.

**Table 1 membranes-15-00113-t001:** Specification of HPNF membranes in the phase 3 NF-RO-MBC pilot system.

Vendor	Model No.	Values
FTS H_2_O	HBR-TFC-4040 (4-inch)	Operating conditions: Maximum Operating Pressure: 1150 psi (80 bar) Maximum Temperature: 113 °F (45 °C) Feedwater pH Limits: 3.0–10.0 Test conditions: NaCl Feed Concentration: 90,000 ppm Applied Pressure: 1050 psi pH Range: 6.5 to 7.0, Temperature: 25 °C Permeate Salinity: 35,000 ppm, Reject Salinity: 96,000 ppm

**Table 2 membranes-15-00113-t002:** Summary of water quality parameters for seawater and RO brine.

Item ^(1)^	Seawater	RO Brine (Feed to HPNF)
pH	8.2 ± 0.15	6.62 ± 0.54
Ca^2+^	0.45 ± 0.08	0.39 ± 0.07
Mg^2+^	1.5 ± 0.15	0.41 ± 0.16
Na^+^	14.2 ± 0.510	27.90 ± 0.30
K^+^	0.42 ± 0.06	1.13 ± 0.04
HCO_3_^−^	0.13 ± 0.01	0.14 ± 0.16
SO_4_^2−^	3.1 ± 0.38	0.28 ± 0.30
Cl^−^	25.00 ± 1.14	45.99 ± 0.29
Br^−^	0.07 ± 0.01	0.21 ± 0.00
TDS	45.00 ± 2.25	76.29 ± 2.03
NaCl/TDS	0.87 ± 0.01	0.97 ± 0.01

^(1)^ Conductivity in [mS/cm], TDS in [g/L], ion compositions in [g/L].

**Table 3 membranes-15-00113-t003:** Design matrix and simulation results for response surface methodology (RSM) on HPNF system.

Run No.	Factors			Responses			
	Feed Flow Rate (m^3^/hr)	Feed TDS (mg/L)	Feed Pressure (bar)	Average Flux (L/m^2^-h)	SEC (kWh/m^3^)	Recovery	Final Brine TDS (mg/L)
1	3.05	80,500	72.5	11.83	6.14	0.7866	208,552
2	3.05	80,500	72.5	11.83	6.14	0.7866	208,552
3	3.3	77,000	70	12.19	6.097	0.7499	183,578
4	3.05	80,500	72.5	11.83	6.14	0.7866	208,552
5	2.8	77,000	75	11.58	5.952	0.8544	245,461
6	2.63	80,500	72.5	10.95	5.883	0.845	240,531
7	3.3	84,000	70	11.84	6.387	0.7141	182,343
8	3.3	84,000	75	12.36	6.51	0.7577	205,062
9	3.05	80,500	72.5	11.83	6.14	0.7866	208,552
10	2.8	84,000	75	11.38	6.188	0.8236	238,476
jj	2.8	77,000	70	11.27	5.781	0.8179	213,829
12	3.05	74,613.7	72.5	12.06	5.924	0.8152	212,293
13	3.05	80,500	76.7	12.15	6.269	0.8187	232,253
14	3.47	80,500	72.5	12.56	6.419	0.7325	185,983
15	3.05	80,500	72.5	11.83	6.14	0.7866	208,552
16	3.05	80,500	68.3	11.44	6.021	0.7503	187,422
17	3.05	80,500	72.5	11.83	6.14	0.7866	208,552
18	3.3	77,000	75	12.66	6.234	0.7917	208,439
19	2.8	84,000	70	11.02	6.028	0.7843	209,646
20	3.05	86,386.3	72.5	11.59	6.365	0.7583	205,973

Conductivity in [mS/cm], TDS in [g/L], ion compositions in [g/L].

**Table 4 membranes-15-00113-t004:** Fit statistics and the regression equations for the RSM models.

Responses	Model F-Value	Predicted R^2^	Adequate Precision	Regression Equation
Flux	38,824	0.9998	716.50	2.7354 + 2.1920 *Q_b_*_,0_ *−* 2.8×10^−5^ *C_b_*_,0_ + 0.069494 *P_f_*_,1_ − 2.9×10^−5^ *Q_b_*_,0_*C_b_*_,0_ + 0.064 *Q_b_*_,0_*P_f_*_,1_ + 1.43 × 10^−6^ *C_b_*_,0_*P_f_*_,1_ − 0.43048 *Q_b_*_,0_^2^ *−* 1.76×10^−10^ *C_b_*_,0_^2^ − 0.00204 *P_f_*_,1_^2^
SEC	341,700	1.0000	2227.56	−0.40953 + 0.33660 *Q_b_*_,0_ + 6.62 × 10^−6^ *C_b_*_,0_ + 0.06117 *P_f_*_,1_ + 0.000012 *Q_b_*_,0_*C_b_*_,0_ − 0.0142 *Q_b_*_,0_*P_f_*_,1_ -3.57 × 10^−7^ *C_b_*_,0_*P_f_*_,1_ + 0.06181 *Q_b_*_,0_^2^ *+* 1.28×10^−10^ *C_b_*_,0_^2^ + 0.000279 *P_f_*_,1_^2^
Recovery	81,057	0.9999	1082.5	1.1394 − 0.28545 *Q_b_*_,0_ *−* 8.00×10^−6^ *C_b_*_,0_ + 0.014026 *P_f_*_,1_ −7.71 × 10^−7^ *Q_b_*_,0_*C_b_*_,0_ + 0.0192 *Q_b_*_,0_*P_f_*_,1_ + 6.57 × 10^−8^ *C_b_*_,0_*P_f_*_,1_ + 0.01226 *Q_b_*_,0_^2^ + 4.83×10^−12^ *C_b_*_,0_^2^ − 0.00012 *P_f_*_,1_^2^
Final brine TDS	19,611.7	0.9996	477.78	39542 − 1.15 × 10^5^ *Q_b_*_,0_ *−* 0.92961 *C_b_*_,0_ + 8475.6 *P_f_*_,1_ + 0.93657 *Q_b_*_,0_*C_b_*_,0_ − 2576.4 *Q_b_*_,0_*P_f_*_,1_ − 0.070628 *C_b_*_,0_*P_f_*_,1_ + 26,544 *Q_b_*_,0_^2^ *+* 1.6404 × 10^−5^ *C_b_*_,0_^2^ + 72.005 *P_f_*_,1_^2^

**Table 5 membranes-15-00113-t005:** Scenarios for multi-criteria improvement.

Scenario	Average Flux *J_avg_* (L/m^2^-h)	SEC (kWh/m^3^)	Recovery *RR*	Final Brine Concentration *C_b, HPNF_* (mg/L)
1	≥11	≤6.5	≥0.75	≥200,000
2	≥11	≤6.0	≥0.80	≥200,000
3	≥12	≤6.5	≥0.75	≥200,000
4	≥11	≤6.5	≥0.75	≥230,000

## Data Availability

The data presented in this study are available on request from the corresponding authors.

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
