# Peer review of "Process Simulation of High-Pressure Nanofiltration (HPNF) for Membrane Brine Concentration (MBC): A Pilot-Scale Case Study"

_membranes, 2025, doi:10.3390/membranes15040113_

Round 1

Reviewer 1 Report

Comments and Suggestions for Authors

The authors presented a mathematical model for high pressure nanofiltration (HPNF) processs for membrane brine concentration (MBC), the research work is interesting and useful, however, some points could be improved:

- Please revise the following sentence in the introduction section: “The High-pressure nanofiltration (HPNF), also known as low-salt rejection reverse osmosis (LSRRO), represents a novel approach to the implementation of MBC. This methodology employs „loose”…”

- The water permeability (A) could be confused with the membrane area (Am), please use a different letter or symbol to define the water permeability. 

- According to Figure 3(b), the model prediction is lower accurate at higher TDS values, could you provide an explanation for this behavior?

Sm,i was not defined

Please summarize the mathematical models reported in literature for HPNF process and highlighted the advantages (or differences) of the presented model. 

- The membranes lifetime is an important parameter, perhaps, it could be included in your analysis, since as the operation pressure and/or feed SDT increase, the membrane lifetime decreases. 

- The abstract and conclusion sections must be improved, are lack and the main results are missed.

- Please include the mathematical models reported in literature for HPNF process and experimental reports for this process. 

- What is the scope of your mathematical model? Is only for the experimental device tested and its experimental conditions or is for wide application?

Author Response

Responses to Reviewer’s Comments

Title: Model development and process simulation of high pressure nanofiltration (HPNF) for membrane brine concentration (MBC)

Reviewer #1

Overall: The authors presented a mathematical model for high pressure nanofiltration (HPNF) processes for membrane brine concentration (MBC), the research work is interesting and useful, however, some points could be improved:

Response:

We appreciate you and the reviewers for your precious time in reviewing our paper and providing valuable comments. It was your valuable and insightful comments that led to possible improvements in the current version. We have carefully considered the comments and tried our best to address every one of them. We hope the manuscript after careful revisions meet your high standards. We welcome further constructive comments if any.

    Below we provide the point-by-point responses. All modifications in the manuscript have been highlighted in red. Thank you again for your helpful comments and our responses are as follows

Comment #1:

Please revise the following sentence in the introduction section: “The High-pressure nanofiltration (HPNF), also known as low-salt rejection reverse osmosis (LSRRO), represents a novel approach to the implementation of MBC. This methodology employs „loose”…”

Response #1:

Thank you very much for your suggestion. The sentence in the introduction has been revised (page 2, marked in red).

Comment #2:

The water permeability (A) could be confused with the membrane area (Am), please use a different letter or symbol to define the water permeability.

Response #2:

Thank you very much for your suggestion. To reflect your suggestion, the symbol for membrane area has been changed from Am to Sm in the revised manuscript.

Comment #3:

According to Figure 3(b), the model prediction is lower accurate at higher TDS values, could you provide an explanation for this behavior?

Response #3:

Thank you very much for your comment. This is probably because the measurement of the TDS in the pilot plant was less accurate than that of the flow rate. The high TDS values were measured in 3rd HPNF stage, which has more uncertainties than the other stages, leading larger fluctuations (see Figure 2(b)). 

In response to your comment, the manuscript has been revised to explain the possible reasons for lower accurate at higher TDS values (page 8-9, marked in red):

“The model prediction is less accurate at higher TDS values. This is probably because the measurement of the TDS in the pilot plant was less accurate than that of the flow rate. The high TDS values were measured in 3rd HPNF stage, which has more uncertainties than the other stages, leading larger fluctuations (see Figure 2(b)).”

Comment #4:

Sm,i was not defined

Response #4:

Thank you very much for catching our mistakes. The definition of Sm,i, which is the membrane area of the ith stage, has been included in the revised manuscript as you suggested. The manuscript has been revised to include the definition of Sm,i as you suggested (page 5, marked in red).

Comment #5:

Please summarize the mathematical models reported in literature for HPNF process and highlighted the advantages (or differences) of the presented model.

Response #5:

Thank you very much for your comments. The following sentences have been included in the revised manuscript to summarize previous models and highlight the advantages of our model (page 2~3, marked in red):

“In the development of HPNF (LSRRO), mathematical modelling and process simulation have become indispensable tools for optimizing operational conditions and membrane configurations [32]. The models for HPNF are used to predict system performance, including flux, permeate quality, and energy efficiency [26]. To elucidate the operating principles of HPNF, a theoretical model based on mass balance was developed and applied to 2-, 3-, and 4-stage HPNF configurations [25]. Membrane transport equations based on a solution-diffusion model were combined with the mass balance equations to elaborate the effect of membrane properties and operating conditions on the performance of a multi-stage HPNF [27]. A method to determine this maximum allowable NaCl retention was developed to show that current RO and NF membranes are not suitable for HPNF [28]. A cost optimization model for HPNF was also developed by combining several sub-models such as membrane stage model, NaCl solution property model and cost model [33]. The performance of LSRRO was theoretically compared with other process configurations such as split-feed counterflow reverse osmosis (SF-CFRO), cascading osmotically mediated reverse osmosis (COMRO), and osmotically assisted reverse osmosis (OARO) [26,34]. The majority of these studies have focused on theoretical analysis, either without experimental data or with only limited data from laboratory-scale tests [25]. Since few models developed based on data collected from real HPNF plant data [28,35], it is difficult to use previous models for practical implementation.”

“The model was established by combining the RO transport models with process mass balance equations. It was calibrated using operational data from a pilot-scale HPNF system and applied for wide application such as process simulation and optimization. Subsequently, a series of HPNF simulation was conducted under various conditions to provide valuable insights into the key factors influencing the process efficiency, including membrane characteristics, operating conditions, and feed water composition. Unlike previous models, our model has been developed based on a real pilot plant for HPNF. Moreover, the model contains only essential equations and is therefore simple, requiring a minimum number of parameters determined by experiments. In addition, the model performs process optimization based on Response Surface Methodology (RSM).”

Comment #6:

The membranes lifetime is an important parameter, perhaps, it could be included in your analysis, since as the operation pressure and/or feed SDT increase, the membrane lifetime decreases.

Response #6:

Thank you very much for your comment. As you pointed out, the membranes lifetime is definitely an important parameter. However, there is no information on the lifetime of HPNF membranes because no long-term test have been conducted so far. Since the operating conditions are quite different between normal NF/RO and HPNF, it is not reasonable to assume that the lifetime of those membranes is the same. Accordingly, this paper could not include the effect of membrane lifetime. Nevertheless, our ongoing study will address this issue in the future. Again, thank you very much for bringing this issue.

In response to your comment, the manuscript has been revised to explain the importance of the membrane lifetime and the related assumption as follows (page 4, marked in red):

“In our model, the water and salt permeabilities (A and B) are assumed to be constant with time. This means that the effect of membrane lifetime is not included. Although this effect is important, it could not be considered because there is no information on the lifetime of HPNF membranes due to the lack of long-term operation study. Accordingly, it should be noted that our current model is applicable to the initial state of HPNF operation.”

Comment #7:

The abstract and conclusion sections must be improved, are lack and the main results are missed.

Response #7:

Thank you very much for your comment. As you suggested, the abstract and conclusions have been improved and included main results in the revised manuscript.

Abstract (marked in red): “Considering the limitations of the pilot plant data, the model showed reasonable accuracy in predicting flux and TDS, with R² values above 0.99. The simulation results demonstrated that an in-crease in feed flow rate improves flux but raises specific energy consumption (SEC) and reduces recovery. In contrast, an increase in feed pressure results in an increased recovery and brine con-centration. Increasing feed TDS decreases flux, recovery, and final brine TDS, and increases SEC.”

Conclusions (marked in red):

  1. A mathematical model for HPNF was developed for MBC and calibrated using pilot plant data. The model showed high accuracy in predicting flux and TDS, with R² values above 0.99, confirming its reliability.
  2. Simulations showed that the lower feed flow rates, higher feed pressures favored recovery and brine concentration. Increasing flow rate and feed pressure improved flux at the expense of energy efficiency. An increase in feed TDS reduces flux, recovery, and final brine TDS, and increases SEC.
  3. Response Surface Methodology (RSM) was used to optimize flux, SEC, recovery, and brine concentration. The F-values for all responses ranges from 81,057 to 341,700. The predicted R² values for all responses are close to 1 and the adequate precision values are all well above the threshold value of 4. These results indicate that the developed models are statistically reliable.
  4. Using the RSM models, the optimum conditions were explored for four typical scenarios of MBC operation. Depending on the criteria in the scenarios, the optimal feed flow rate and pressures vary. For instance, the feed flow rate should be low and pressure should be intermediate when high recovery (≥ 0.80) and low SEC (≤ 6.0 kWh/m3) are required (scenario 2). On the other hand, the feed flow rate and pres-sure should be high to achieve high flux (≥ 12 L/m2-hr) is required (scenario 3). This suggests the importance of systematic process optimization for MBC.

Comment #8:

Please include the mathematical models reported in literature for HPNF process and experimental reports for this process.

Response #8:

Thank you very much for your comment. Although there are several papers on mathematical models, few papers on experimental reports are available except for our previous work. In response to your comment, the manuscript has been revised to include  (page 2, marked in red):    

“The models for HPNF are used to predict system performance, including flux, permeate quality, and energy efficiency [26]. To elucidate the operating principles of HPNF, a theoretical model based on mass balance was developed and applied to 2-, 3-, and 4-stage HPNF configurations [25]. Membrane transport equations based on a solution-diffusion model were combined with the mass balance equations to elaborate the effect of membrane properties and operating conditions on the performance of a multi-stage HPNF [27]. A method to determine this maximum allowable NaCl retention was developed to show that current RO and NF membranes are not suitable for HPNF [28]. A cost optimization model for HPNF was also developed by combining several sub-models such as membrane stage model, NaCl solution property model and cost model [33]. The performance of LSRRO was theoretically compared with other process configurations such as split-feed counterflow reverse osmosis (SF-CFRO), cascading osmotically mediated reverse osmosis (COMRO), and osmotically assisted reverse osmosis (OARO) [26,34].”

Comment #9:

What is the scope of your mathematical model? Is only for the experimental device tested and its experimental conditions or is for wide application?

Response #9:

The scope of our model is not only for our experimental system but also for wider application. The equations used in our model are not limited to our experimental system and thus may be used for other pilot- and full-scale HPNF processes. 

In response to your comment, the manuscript has been revised to clarify this (page 2, marked in red):  

“The model was established by combining the RO transport models with process mass balance equations. It was calibrated using operational data from a pilot-scale HPNF system and applied for wide application such as process simulation and optimization.”

Reviewer 2 Report

Comments and Suggestions for Authors

The article explores the use of high-pressure nanofiltration (HPNF) technology for the concentration of brines, aiming to provide an efficient alternative to conventional methods like mechanical vapor compression (MVC). The study focuses on developing and validating a model to predict the performance of HPNF systems, based on pilot-scale experimental data.

Comments.

1.      The introduction does not sufficiently explain the advantages of HPNF compared to alternative technologies like mechanical vapor compression (MVC). A more detailed discussion of its competitive benefits is recommended.

2.      The description of pilot project data is limited. More information on data reproducibility and experimental conditions is needed.

3.      While the correlation between the model and pilot project data is strong (R² > 0.99), there is no discussion of model errors or their potential impact on predictions.

4.      The model is focused solely on brine concentration without considering other contaminants, such as organic compounds, reducing its versatility.

  1. Insufficient information is provided on the scalability of the proposed solution for large-scale industrial applications.
  2. Although specific energy consumption (SEC) is discussed, more comparative data on HPNF efficiency relative to conventional methods under real-world conditions is needed.
  3. The Response Surface Methodology (RSM) is well-described, but there is a lack of in-depth analysis of variable interactions. Adding graphical representations of these interactions would be beneficial.
  4. The article lacks an analysis of environmental impacts, such as the carbon footprint of HPNF compared to traditional technologies.
  5. Limitations on pH range, temperature, and salinity are mentioned, but it is unclear how the system adapts to changing input conditions.
  6. Certain terms (e.g., "loose RO") require more detailed explanations for readers unfamiliar with the field.
  7. The article does not include a cost-of-ownership analysis or economic feasibility evaluation of the proposed technology.
  8. Table 1 lists membrane characteristics, but data on membrane durability under operational conditions is missing.
  9. Only recent studies are cited. Including a historical perspective and the development of HPNF would provide valuable context.
  10. Some graphs (e.g., Figures 3 and 6) are not sufficiently clear. Improving their readability is necessary.
  11. There is no information about the software and computational resources used for the simulations.

These comments aim to enhance the quality of the paper, making it more comprehensive and valuable for researchers and practitioners.

Author Response

Responses to Reviewer’s Comments

Title: Model development and process simulation of high pressure nanofiltration (HPNF) for membrane brine concentration (MBC)

Reviewer #2

Overall:

The article explores the use of high-pressure nanofiltration (HPNF) technology for the concentration of brines, aiming to provide an efficient alternative to conventional methods like mechanical vapor compression (MVC). The study focuses on developing and validating a model to predict the performance of HPNF systems, based on pilot-scale experimental data.

Response:

Thank you very much for reviewing our paper. Below we provide the point-by-point responses. All modifications in the manuscript have been highlighted in red. Thank you again for your helpful comments and our responses are as follows.

Comment #1:

The introduction does not sufficiently explain the advantages of HPNF compared to alternative technologies like mechanical vapor compression (MVC). A more detailed discussion of its competitive benefits is recommended.

Response #1:

Thank you very much for your suggestion. According to previous study, the SEC of 2-Stage MVC was reported to 24 kWh/m3 while that of HPNF MBC was found to be 6~8 kWh/m3. Besides, there are many other advantages of MBC over MVC. A more detailed discussion of the competitive benefits of HPNF compared with MVC has been included in the revised manuscript as you suggested (page 2, marked in red):

“For example, HPNF can achieve specific energy consumption (SEC) as low as 2.4 to 8.0 kWh/m³, significantly lower than traditional thermal methods such as mechanical vapor compression (MVC), which consumes 20-25 kWh/m³ [25]. Besides, there are many other advantages of MBC over MVC [21].”

Comment #2:

The description of pilot project data is limited. More information on data reproducibility and experimental conditions is needed.

Response #2:

Thank you very much for your comment. The detailed information on the pilot plant was reported by our previous research paper. The data used for model calibration was previously checked for confirming its reproducibility. In response to your comment, the manuscript has been revised to discuss it (page 6, marked in red):

“More information on the pilot plant, including the operating conditions and data re-producibility, is available in our previous study [24].”

Comment #3:

While the correlation between the model and pilot project data is strong (R² > 0.99), there is no discussion of model errors or their potential impact on predictions.

Response #3:

Thank you very much for your comment. As you pointed out, high R2 values do not always guarantee that the model is useful. In response to your comment, the manuscript has been revised to discuss model errors or their potential impact on predictions (page 8-9, marked in red):

“The model prediction is less accurate at higher TDS values. This is probably because the measurement of the TDS in the pilot plant was less accurate than that of the flow rate. The high TDS values were measured in 3rd HPNF stage, which has more uncertainties than the other stages, leading larger fluctuations (see Figure 2(b)).”

Comment #4:

The model is focused solely on brine concentration without considering other contaminants, such as organic compounds, reducing its versatility.

Response #4:

Thank you very much for pointing out the limitation of our current study. As you pointed out, it is important to consider other contaminants, such as organic compounds. Nevertheless, our current study could not include it because there was no measured data on other contaminants available in our pilot plant. We will further develop other model in the future when the data becomes available. Besides, the feed water of MBC processes is pretreated by NF, which can remove most organic compounds (Desalination, Volume 597, 15 March 2025, 118308). In such cases, our model in this study is still meaningful.

In response to your comment, the manuscript has been revised to discuss the limitation of the model and the use of NF for pretreatment (page 4, marked in red):

“The model is focused on brine concentration without considering other contaminants, such as organic compounds. Although it may reduce its versatility, it could not be included because of lack of data on the concentrations of such contaminants in the pilot plant. Besides, the feed water of MBC processes is pretreated by NF, which can remove most organic compounds [21,23].”

Comment #5:

Insufficient information is provided on the scalability of the proposed solution for large-scale industrial applications.

Response #5:

Thank you very much for your comment. As you pointed out, on the scalability for large-scale industrial applications is important. In response to your comment, the manuscript has been revised to discuss this issue (page 2, marked in red):

“The scalability of HPNF for large-scale industrial applications was discussed in our previous study [24].”

Comment #6:

Although specific energy consumption (SEC) is discussed, more comparative data on HPNF efficiency relative to conventional methods under real-world conditions is needed.

Response #6:

Thank you very much for your comment. Comparison of HPNF with conventional methods was performed in our previous research paper (Desalination, Volume 597, 15 March 2025, 118308). In response to your comment, the manuscript has been revised to discuss this issue (marked in red):

“This process is particularly useful for economically reducing the volume of brine, thereby recovering more fresh water and leaving behind a more concentrated salt solution for potential recovery [29]. For example, the final NaCl concentration by RO-electrodialysis (ED) was higher (244 g/L) than that of the NF-SWRO-HPNF (225) but its energy consumption is 30~44% higher [24,30]. Other ED systems reported a lower NaCl concentration with a higher energy consumption [31].”

Comment #7:

The Response Surface Methodology (RSM) is well-described, but there is a lack of in-depth analysis of variable interactions. Adding graphical representations of these interactions would be beneficial.

Response #7:

Thank you very much for your comment. Graphical representations of variable interactions have been included as a new figure (Figure 8) in the revised manuscript as you suggested.  

Comment #8:

The article lacks an analysis of environmental impacts, such as the carbon footprint of HPNF compared to traditional technologies.

Response #8:

Thank you very much for your suggestion. Needless to say, environmental impacts, such as the carbon footprint are important and should not be missed. Compared with thermal process for brine concentration, HPNF significantly reduces carbon footprint. It was reported that total carbon footprints of multi-stage flash (MSF), multi-effect distillation (MED), and SWRO are 18.0, 14.7, and 4.7 kg CO2-eq. Assuming that the carbon emission of membrane processes is almost proportional to its energy consumption, HPNF may have approximately 2~2.5 times higher carbon footprint than SWRO, which corresponds to 9~14 kg CO2-eq. Nevertheless, it is necessary to conduct more rigorous analysis of carbon footprint of HPNF in comparison with other competing techniques. In response to your comment, the manuscript has been revised to discuss this issue (page 11, marked in red):

“According to these results, the SEC of HPNF ranges from 5.8 kWh/m3 to 6.8 kWh/m3, which is significantly lower than those of thermal desalination techniques applicable for brine concentration. For example, the total carbon footprints of multi-stage flash (MSF), multi-effect distillation (MED), and SWRO were reported to be 18.0, 14.7, and 4.7 kg CO2-eq, respectively. Assuming that the carbon emission of membrane processes is almost proportional to their energy consumption, and that HPNF may have about 2~2.5 times higher SEC than conventional SWRO, the carbon footprint of HPNF is about 9~14 kg CO2-eq. Nevertheless, it is necessary to conduct more rigorous analysis of the carbon footprint of HPNF in comparison with other competing techniques.”

Comment #9:

Limitations on pH range, temperature, and salinity are mentioned, but it is unclear how the system adapts to changing input conditions.

Response #9:

Thank you very much for your comment. Since details on the pilot plant operations have been reported in our previous research paper (Desalination, Volume 597, 15 March 2025, 118308), our original manuscript does not include them. In response to your comment, the manuscript has been revised to refer this paper and provide more information on how the system adapts to changing input conditions (page 6, marked in red):

“Each stage operated in constant pressure mode, which was done by regulating the pressure of the HPNF feed pump. Regardless of the inlet pressure, the final pressure was set by the HPNF feed pump in the pilot plant. The booster pump pressures were set to be the same as the feed pump pressure. The process configuration of the HPNF stages is illustrated in Figure 1. More information on the pilot plant, including the operating conditions and data reproducibility, is available in our previous study [23].”

Comment #10:

Certain terms (e.g., "loose RO") require more detailed explanations for readers unfamiliar with the field.

Response #10:

Thank you very much for your comment. More detailed explanations for “loose RO” and other terms have been included in the revised manuscript as you suggested (page 2, marked in red):

“The High-pressure nanofiltration (HPNF), also known as low-salt rejection reverse osmosis (LSRRO), represents a novel approach to the implementation of MBC [25-27]. This methodology employs “loose”RO or NF membranes to treat highly saline water at elevated hydraulic pressures. These membranes with a lower salt retention than conventional RO membranes can be used to reduce the required hydraulic pressure (transmembrane pressure) for the concentration process [28].”

Comment #11:

The article does not include a cost-of-ownership analysis or economic feasibility evaluation of the proposed technology.

Response #11:

Thank you very much for your comment. The current manuscript covers process simulation and optimization for MBC. Although economic feasibility is important, it is beyond the scope of this manuscript. A separate research paper will be needed to conduct techno-economic analysis of MBC.

In response to your comment, the manuscript has been revised to highlight the importance of economic feasibility evaluation and mention the limitations of our current study (page 17, marked in red).

“The current manuscript does not include a cost-of-ownership analysis or economic feasibility evaluation of HPNF MBC. This is because it is beyond the scope of this work. A techno-economic analysis results for the NF-RO-MBC was conducted in our previous work [12] although a different MBC process was considered. Further work will be required to conduct an economic analysis of the HPNF MBC.”

Comment #12:

Table 1 lists membrane characteristics, but data on membrane durability under operational conditions is missing.

Response #12:

Thank you very much for your comment. The data on membrane durability is important. According to the manufacturer, the maximum operating pressure is 1150 psi (80 bar) and the maximum temperature is 113°F (45°C). Other information on membrane durability is not available.

In response to your comment, the manuscript has been revised to briefly mention the available information on the membrane durability (page 6, marked in red):

“According to the manufacturer, the maximum operating pressure is 1150 psi (80 bar) and the maximum temperature is 113°F (45°C). Other information on membrane durability is not available. The specifications of the HPNF membranes are summarized in Table 1.”

Comment #13:

Only recent studies are cited. Including a historical perspective and the development of HPNF would provide valuable context.

Response #12:

Thank you very much for your comment. However, HPNF (or LSRRO) is an emerging process and the first journal paper was published in 2020 (e.g. Water Research Volume 170, 1 March 2020, 115317).

In response to your comment, the manuscript has been revised to include journal papers to a historical perspective:

  1. Wang, Z.; Deshmukh, A.; Du, Y.; Elimelech, M. Minimal and zero liquid discharge with reverse osmosis using low-salt-rejection membranes. Water Research 2020, 170, 115317, doi:https://doi.org/10.1016/j.watres.2019.115317.
  2. Atia, A.A.; Yip, N.Y.; Fthenakis, V. Pathways for minimal and zero liquid discharge with enhanced reverse osmosis technologies: Module-scale modeling and techno-economic assessment. Desalination 2021, 509, 115069, doi:https://doi.org/10.1016/j.desal.2021.115069.
  3. Du, Y.; Wang, Z.; Cooper, N.J.; Gilron, J.; Elimelech, M. Module-scale analysis of low-salt-rejection reverse osmosis: Design guidelines and system performance. Water Research 2022, 209, 117936, doi:https://doi.org/10.1016/j.watres.2021.117936.

Comment #14:

Some graphs (e.g., Figures 3 and 6) are not sufficiently clear. Improving their readability is necessary.

Response #14:

Thank you very much for pointing out the graph quality issues. Figure 3 and 6 were updated to improve the readability. Other figures (Figure 1, Figure 7, and Figure 8) were also improved to have sufficient quality.

Comment #15:

There is no information about the software and computational resources used for the simulations.

Response #15:

A commercial program, Engineering Equation Solver (EES), was used for the simulation. A desktop PC (13th Gen Intel® I7-13700K, 128 GB RAM) was used to run the program. Details on the software and computational resources have been included in the revised manuscript as you suggested (page 6, marked in red).

“The above equations were simultaneously solved using the EES software (F-Chart, U.S.A.). A desktop PC (13th Gen Intel® I7-13700K, 128 GB RAM) was used to run the program.”

Comment #16:

These comments aim to enhance the quality of the paper, making it more comprehensive and valuable for researchers and practitioners.

Response #16:

We appreciate you and the reviewers for your precious time in reviewing our paper and providing valuable comments. It was your valuable and insightful comments that led to possible improvements in the current version. We have carefully considered the comments and tried our best to address every one of them. We hope the manuscript after careful revisions meet your high standards. We welcome further constructive comments if any.

Reviewer 3 Report

Comments and Suggestions for Authors

Comments on the manuscript Membranes-3429795 entitled “Model development and process simulation of high pressure nanofiltration (HPNF) for membrane brine concentration (MBC)” This manuscript deals with model development and simulation of a process (designated as HPNF) for treatment of RO-concentrate (from sea-water) to recover additional desalinated water. The authors aim to make a contribution of some practical interest, apparently related to improved sustainability of the sea-water desalination. However, this reviewer has the following serious concerns, mainly regarding the approach taken, the techniques employed, the quality of presentation and the value of the results obtained.

1. Scope, approach and substance of the work
1.1 The authors deal only with a HPNF processing scheme (Figure 1), which is part of a general desalination process involving a) seawater NF pretreatment, b) RO desalination, c) HPNF section treating the RO concentrate. Curiously, the work is restricted to HPNF and essentially nothing is said about the first two (a, b) sections, even though they are intimately related and the permeate from HPNF is recycled to RO for final treatment. For instance:
i) It is totally unclear how/if the recovery and the concentrate of the first/NF section are taken into account in the attempts to model/optimize the system. ii) There is no mention of the specific energy consumption (SEC) of the entire membrane plant/system; dealing only with the SEC of HPNF stages and ignoring the SEC of the entire process/plant (i.e. NF/RO/HPNF) is questionable, especially when optimization of the system is pursued.

1.2 A fixed design of HPNF section, comprised of 3 sequential stages with repeated permeate recycling, was considered with no explanation. This is a serious limitation.

1.3 A very crude and unclear “modeling” of the HPNF process is presented, based on simple mass balances. In fact, nothing is said about the three membrane stages of the HPNF section (Fig. 1); e.g. i) whether the feed and permeate flow rates used in calculations correspond to existing/realistic SWM modules/vessels; ii) whether each stage operates “in constant flux”, or “constant pressure” mode, or how. These are significant omissions, impacting on the value of the results obtained.

1.4 Further on the above, the approach to validate/adjust? the model using pilot-plant data is also unclear. These data are inadequately described, regarding mode of pilot-plant operation and arrangement of used SWM modules. For instance, it is merely stated (section 3.1) that “The numbers of the elements in the 1st, 2nd, and 3rd HPNF stages were 12, 8, 4, respectively. The applied pressure was 70 bar”. Questions:
i) Were these elements arranged in series, in parallel vessels, or how?
ii) Where exactly was the 70 bar pressure applied? This pressure should be clearly marked with symbols of Fig. 1, considering that feed pressure in each stage was apparently different.

1.5 The OFAT approach to deal with the data is considered merely indicative and of little (if any value) for process simulation. It is used here with inadequate explanation of its obvious limitations.

1.6 No comparison whatsoever was made with literature results, for any quantity dealt with in this manuscript.

1.7 In general, the authors deal in a simplistic manner with a significant and quite difficult problem; i.e. the overall sustainability of the seawater desalination. By isolating to model and “optimize”, in the manner implemented here, only the membrane brine concentration (MBC) by HPNF, is considered inadequate and of little value. However, of prime interest these days is the optimization and SEC of the entire plant as well as the valorization of all the effluents/concentrates (in this case from both NF- 1st section and the HPNF process).

2. Problematic quality of presentation
2.1 In addition to the above issues, the manuscript needs considerable attention regarding organization of material and clarity of presentation. The paper is difficult to read, particularly the Results section. A few examples for corrections and/or improvements follow:
i) The definition of symbols with many subscripts in the Figures and Tables (particularly in Figure 1) needs attention. In most Figures these symbols are very small and illegible.
ii) Sections 2.2 and 2.3 have exactly the same (confusing) title. iii) Need to check Equation (12); the quantity Pf,1 (2nd term in the nominator) seems to be incorrect. iv) Why a quite low pump efficiency (0,70?) was used to calculate SEC? Moreover, it should be noted that the computed SEC of HPNF process is much larger than that of currently operating RO plants. v) In Table 5, the parameters listed need clear definition, using the same symbols as those in the process diagram (Figure 1). vi) Figure 8 presenting results of the ‘multi-criteria optimization’ is of little use in the manner presented and with totally illegible parameter values. vii) Considering all the aforementioned issues and limitations, it is inappropriate to state in Abstract and Conclusions that ‘The model showed high accuracy in predicting flux, etc…’ In summary, the approach taken and the value of reported results are questionable; therefore, this manuscript is not considered fit for publication in a scientific journal.

Author Response

Responses to Reviewer’s Comments

Title: Model development and process simulation of high pressure nanofiltration (HPNF) for membrane brine concentration (MBC)

Reviewer #3

Overall:

Comments on the manuscript Membranes-3429795 entitled “Model development and process simulation of high pressure nanofiltration (HPNF) for membrane brine concentration (MBC)” This manuscript deals with model development and simulation of a process (designated as HPNF) for treatment of RO-concentrate (from sea-water) to recover additional desalinated water. The authors aim to make a contribution of some practical interest, apparently related to improved sustainability of the sea-water desalination. However, this reviewer has the following serious concerns, mainly regarding the approach taken, the techniques employed, the quality of presentation and the value of the results obtained.

Response:

We appreciate you and the reviewers for your precious time in reviewing our paper and providing valuable comments. It was your valuable and insightful comments that led to possible improvements in the current version. We have carefully considered the comments and tried our best to address every one of them. We hope the manuscript after careful revisions meet your high standards. We welcome further constructive comments if any.

Below we provide the point-by-point responses. All modifications in the manuscript have been highlighted in red. Thank you again for your helpful comments and our responses are as follows

Comment #1:

  1. Scope, approach and substance of the work
    1.1 The authors deal only with a HPNF processing scheme (Figure 1), which is part of a general desalination process involving a) seawater NF pretreatment, b) RO desalination, c) HPNF section treating the RO concentrate. Curiously, the work is restricted to HPNF and essentially nothing is said about the first two (a, b) sections, even though they are intimately related and the permeate from HPNF is recycled to RO for final treatment. For instance:
    i) It is totally unclear how/if the recovery and the concentrate of the first/NF section are taken into account in the attempts to model/optimize the system. ii) There is no mention of the specific energy consumption (SEC) of the entire membrane plant/system; dealing only with the SEC of HPNF stages and ignoring the SEC of the entire process/plant (i.e. NF/RO/HPNF) is questionable, especially when optimization of the system is pursued.

Response #1:

We sincerely thank you for your insightful feedback and constructive critique. It is our mistake not to cite our previous research paper (Desalination Volume 597, 15 March 2025, 118308) that includes all the details of the target system. Since this paper is a follow-up of our previous paper, we did not include the explanations and justifications of NF/SWRO/HPNF system. The manuscript has been revised to cite our previous paper and provide brief explanations on this system (page 2, marked in red). 

“The process configuration, which consists of nanofiltration (NF), SWRO, and HPNF, was selected for model development. Our previous study analyzed the applicability and effectiveness of this system for brine concentration”

As you pointed out, the interconnected nature of the whole desalination process (NF-SWRO-HPNF) is important. Our study focuses on the HPNF stage as its primary novelty lies in optimizing the treatment of RO concentrate, which poses significant technical and sustainability challenges. However, we acknowledge that the broader system context is critical and have addressed this as follows:

  • The first NF and SWRO stages were treated as boundary conditions in this work. Accordingly, the current mode is not intended to optimize the entire NF-SWRO-HPNF system. In most cases, the recoveries of NF and SWRO stages have rather narrow ranges (e.g., NF recovery = 70~80%, RO recovery = 40~50%) and thus cannot be easily adjusted. Nevertheless, future work will explicitly integrate variable NF/RO performance into a holistic optimization framework.
  • We agree that evaluating the SEC of the entire system (NF-SWRO-HPNF) is essential for practical implementation. However, this study prioritizes the HPNF stage because the energy consumption of HPNF is far higher (6~8 kWh/m3) than those of conventional NF (typically 1~3 kWh/m3) and RO (typically 2.5~4 kWh/m3) stages, making it the dominant contributor to system-wide SEC. Nevertheless, future work will be needed to optimize the SEC of the whole system by a holistic approach.

In response to your comment, the manuscript has been revised to add clarification regarding the focus of this study and the need for holistic optimization (page 17, marked in red):

“Since this study focuses on the HPNF stage, the interconnected nature of the entire desalination process (NF-SWRO-HPNF) is not considered. The limitations of this study can be summarized as follows:

  1. Instead of the considering the recovery of the whole process, only the recovery of the HPNF was considered in this work. The first NF and SWRO stages were treated as boundary conditions. Accordingly, the current mode is not intended to optimize the entire NF-SWRO-HPNF system. Future work will explicitly integrate variable NF/RO performance into a holistic optimization framework.
  2. The SEC of the overall process was not included in the model. This study prioritiz-es the HPNF stage because the energy consumption of HPNF is much higher than that of SWRO and NF. Nevertheless, future work is needed to optimize the SEC of the whole system by a holistic approach.”

Comment #2:

1.2 A fixed design of HPNF section, comprised of 3 sequential stages with repeated permeate recycling, was considered with no explanation. This is a serious limitation.

Response #2:

We sincerely thank you for your valuable comment. As you pointed out, our simulation only considered the 3-stage HPNF. This is because this design was found to be an appropriate configuration for MBC. Details on this justification has been reported in our previous research paper.

In response to your comment, the manuscript has been revised to justify the selection of the 3-stage HPNF system by citing our previous research paper (page 2, marked in red):

“The process configuration, which consists of NF, SWRO, and HPNF, was selected for model development. Our previous study analyzed the applicability and effectiveness of this system for brine concentration [24].”

Comment #3:
1.3 A very crude and unclear “modeling” of the HPNF process is presented, based on simple mass balances. In fact, nothing is said about the three membrane stages of the HPNF section (Fig. 1); e.g. i) whether the feed and permeate flow rates used in calculations correspond to existing/realistic SWM modules/vessels; ii) whether each stage operates “in constant flux”, or “constant pressure” mode, or how. These are significant omissions, impacting on the value of the results obtained.

Response #3:

We sincerely thank you for your valuable comment. It is clearly our mistake to omit important information about the membrane system. The detailed information is as follows:

  • It was confirmed that the feed and permeate flow rates used in calculations correspond to existing/realistic SWM modules/vessels. All the conditions for the simulation were selected based on the conditions of the real pilot plant.
  • Each stage operated in constant pressure mode. This was done by regulating the pressure of the feed pump. The booster pump pressures were set to be the same as the feed pump pressure.

In response to your comment, the manuscript has been revised to include this information and cite our previous research paper that contains additional information on the pilot plant:  

(Page 9, marked in red) “Before the simulation, it was confirmed that the feed and permeate flow rates used in calculations correspond to existing/realistic 4-inch spiral modules/vessels. All the conditions for the simulation were selected based on the conditions of the real pilot plant.”

(Page 6, marked in red) “Each stage operated in constant pressure mode, which was done by regulating the pressure of the HPNF feed pump. The booster pump pressures were set to be the same as the feed pump pressure.”

Comment #4:

1.4 Further on the above, the approach to validate/adjust? the model using pilot-plant data is also unclear. These data are inadequately described, regarding mode of pilot-plant operation and arrangement of used SWM modules. For instance, it is merely stated (section 3.1) that “The numbers of the elements in the 1st, 2nd, and 3rd HPNF stages were 12, 8, 4, respectively. The applied pressure was 70 bar”. Questions:
i) Were these elements arranged in series, in parallel vessels, or how?
ii) Where exactly was the 70 bar pressure applied? This pressure should be clearly marked with symbols of Fig. 1, considering that feed pressure in each stage was apparently different.

Response #4:

We sincerely appreciate your valuable comment. It is clearly our mistake to omit important information about the membrane system. The detailed information is as follows:

  • Each vessel contains four elements. Accordingly, the numbers of vessels in the 1st, 2nd, and 3rd HPNF stages are 3, 2, and 1 respectively.
  • In each stage, the vessels were arranged in parallel.
  • The pressure was applied using the feed pump, which is connected to the 1st HPNF stage. Regardless of the inlet pressure, the final pressure was set to 70 bar in the pilot plant. For instance, if the pressure of the SWRO brine is 60 bar, the HPNF feed pump increases the pressure from 60 bar to 70 bar. Since the model does not include the SWRO stage, the effect of SWRO pressure was not considered.

In response to your comment, the manuscript has been revised to include this information and cite our previous research paper that contains additional information on the pilot plant (page 6, marked in red):

“Each vessel contains four elements. Accordingly, the numbers of vessels in the 1st, 2nd, and 3rd HPNF stages are 3, 2, and 1 respectively. The vessels in each stage were arranged in parallel. The applied pressure was 70 bar. Each stage operated in constant pressure mode, which was done by regulating the pressure of the HPNF feed pump. Regardless of the inlet pressure, the final pressure was set to 70 bar by the HPNF feed pump in the pilot plant. The booster pump pressures were set to be the same as the feed pump pressure. The specifications of the HPNF membranes are summarized in Table 1. The process configuration of the HPNF stages is illustrated in Figure 1. More information on the pilot plant is available in our previous study [24].”

Comment #5:

1.5 The OFAT approach to deal with the data is considered merely indicative and of little (if any value) for process simulation. It is used here with inadequate explanation of its obvious limitations.

Response #5:

We sincerely thank you for your valuable comment. The traditional one-factor-at-a-time (OFAT) approach can serve the purpose of coarse estimation of the optimum levels, it is time-consuming and cannot evaluate the interactions among factors and responses.

In response to your comment, the manuscript has been revised to explain the drawbacks of the OFAT approach and to provide a reference (page 11, marked in red):   

“Although the OFAT approach is beneficial for understanding the effect of operating variables on process performance, it is inadequate for analyzing the interactions the interactions among factors and responses and time-consuming to find the optimum conditions [40].”

Comment #6:

1.6 No comparison whatsoever was made with literature results, for any quantity dealt with in this manuscript.

Response #6:

We sincerely appreciate your valuable comment. As far as we know, there is only one pilot plant study on HPNF processes in literature, which is our previous research paper (Desalination Volume 597, 15 March 2025, 118308). Since we used the data collected from the same pilot plant, the results were compared and found to be consistent. The comparison of this work with other works done in lab-scale was not carried out because our model intends to design and optimize either pilot- or full-scale plants. We will do it in the future when the relevant works are reported.

Comment #7:

1.7 In general, the authors deal in a simplistic manner with a significant and quite difficult problem; i.e. the overall sustainability of the seawater desalination. By isolating to model and “optimize”, in the manner implemented here, only the membrane brine concentration (MBC) by HPNF, is considered inadequate and of little value. However, of prime interest these days is the optimization and SEC of the entire plant as well as the valorization of all the effluents/concentrates (in this case from both NF- 1st section and the HPNF process).

Response #7:
We sincerely appreciate your critical comment and acknowledge the importance of considering the overall sustainability of seawater desalination. Our study aims to contribute to this broader objective by focusing on MBC process using HPNF. While we recognize that a comprehensive optimization of the entire desalination plant, including energy consumption (SEC) and effluent valorization, is crucial, our work is positioned as a focused step within this larger framework. 

The rationale for isolating the MBC process in our study is twofold: (1) To provide a theoretical evaluation of HPNF performance in brine concentration, which remains a key challenge in enhancing overall desalination efficiency; and (2) To establish a foundation for integrating MBC into broader system-level optimizations in future studies. We acknowledge that a holistic approach incorporating both SWRO and HPNF, along with the valorization of concentrates, is of significant interest, and we aim to address this in follow-up research.

To further align our study with the broader sustainability objectives, we have made additional clarifications in the manuscript regarding the role of MBC in overall desalination plant optimization. Furthermore, we have included a discussion on how our findings can contribute to future integrated modeling efforts that encompass full plant-wide energy and resource optimization.

We appreciate your valuable insights and believe that our revised manuscript clarifies the importance of our work in the broader context of sustainable seawater desalination (page 3, marked in red): 

“Our study aims to contribute to this broader objective by focusing on MBC process using HPNF. While comprehensive optimization of the entire desalination plant, including energy consumption (SEC) and effluent valorization, is crucial, this study is positioned as a focused step within this larger framework. The rationale for isolating the MBC process in our study is twofold: (1) To provide a theoretical evaluation of HPNF performance in brine concentration, which remains a key challenge in enhancing overall desalination efficiency; and (2) To establish a foundation for integrating MBC into broader system-level optimizations in future studies.”

Comment #8:
2. Problematic quality of presentation
2.1 In addition to the above issues, the manuscript needs considerable attention regarding organization of material and clarity of presentation. The paper is difficult to read, particularly the Results section. A few examples for corrections and/or improvements follow:

Response #8:
We sincerely thank you for pointing out the issues related to the quality of the presentation. We paid considerable attention to improve the organization of material and clarity of presentation. We also attempted to resolve the readability issue that you pointed out. Below we provide the point-by-point responses to your valuable comments.

Comment #9:
i) The definition of symbols with many subscripts in the Figures and Tables (particularly in Figure 1) needs attention. In most Figures these symbols are very small and illegible.

Response #9:
We sincerely thank you for your comments. The symbols in Figure 1 and other figures have been improved in the revised manuscript as you suggested.

Comment #10:

  1. ii) Sections 2.2 and 2.3 have exactly the same (confusing) title.

Response #10:
We sincerely thank you for catching the errors. The manuscript was revised to correct them as you pointed out:

“2.2. Membrane transport equations for HPNF”

“2.3. Mass balance in HPNF”

Comment #11:

iii) Need to check Equation (12); the quantity Pf,1 (2nd term in the nominator) seems to be incorrect.

Response #11:
We sincerely thank you for checking the equation. The equation (12) is not incorrect. The permeate from the 2nd stage (Qp,2) should be pressurized to the applied pressure of the 1st stage (Pf,1). Similarly, the permeate from the 3rd stage (Qp,3) should be pressurized to the applied pressure of the 2nd stage (Pf,2). In addition, the feed to the 1st stage (Qf,1) should be pressurized to the applied pressure of the 1st stage (Pf,1). Figure 1 has been revised to clarify this.

Comment #12:

  1. iv) Why a quite low pump efficiency (0,70?) was used to calculate SEC? Moreover, it should be noted that the computed SEC of HPNF process is much larger than that of currently operating RO plants.

Response #12:
The pump efficiency in the model was set to be similar to that in the pilot plant. The efficiency is relatively low because the pump capacity was about 100~125 m3/day. The efficiencies of such small pumps are much lower than those of large pumps used in large-scale desalination plants.

In response to your comment, the manuscript has been revised to explain why the pump efficiency was set to 0.70 (page 9, marked in red):

“The pump efficiency in the model was set to be similar to that in the pilot plant [24]. The efficiency is relatively low because the pump capacity was about 100~125 m3/day. The efficiencies of such small pumps are much lower than those of large pumps used in large-scale desalination plants (~0.85).”

Comment #13:

  1. v) In Table 5, the parameters listed need clear definition, using the same symbols as those in the process diagram (Figure 1).

Response #13:
We sincerely appreciate your suggestion. The manuscript was revised to provide clear definitions for parameters in Table 5 (page 13, marked in red):  

“The average flux (Javg), SEC, and recovery (RR) are defined in eq.(10), eq.(12), and eq.(13). The final brine concentration (cb,HPNF) is identical to cb,3 as shown in Figure 1.”

Comment #14:

  1. vi) Figure 8 presenting results of the ‘multi-criteria optimization’ is of little use in the manner presented and with totally illegible parameter values.

Response #14:

We sincerely thank you for pointing out this issue. The manuscript was revised to revise the Figure 8 (Figure 9 in revised manuscript) as you suggested. Thank you again.

Comment #15:

vii) Considering all the aforementioned issues and limitations, it is inappropriate to state in Abstract and Conclusions that ‘The model showed high accuracy in predicting flux, etc…’

Response #15:

We sincerely appreciate for your comment. As you suggested, the statement in the Abstract and Conclusions has been changed to ‘Considering the limitation of data in the pilot plant, the model showed reasonable accuracy in predicting flux and TDS, …’ If this statement is not yet appropriate, we will revise it again.

Comment #16:

In summary, the approach taken and the value of reported results are questionable; therefore, this manuscript is not considered fit for publication in a scientific journal.

Response #16:

We agree that our previous manuscript is not suitable for publication in this journal. However, we have carefully considered the comments and have done our best to address each of them. We hope that after careful revision, the manuscript will meet your high standards. We believe that your comments have significantly improved the quality of our manuscript. We would like to reconsider your decision after reviewing the revised manuscript. If you have any further comments on the revised manuscript, we will do our best to revise the manuscript again.

Round 2

Reviewer 1 Report

Comments and Suggestions for Authors

All comments were attended. I recommend for publication in present form.

Author Response

Comments 1: All comments were attended. I recommend for publication in present form.

Response 1: Thank you very much for your time and valuable insights. Your recommendation for publication is deeply appreciated.

Reviewer 3 Report

Comments and Suggestions for Authors

Attached comments

Author Response

Response to Reviewer 3 Comments

1. Summary

2. Questions for General Evaluation

Reviewer’s Evaluation

Response and Revisions

Does the introduction provide sufficient background and include all relevant references?

Can be improved

Improved (see below)

Is the research design appropriate?

Must be improved

Improved (see below)

Are the methods adequately described?

Can be improved

Improved (see below)

Are the results clearly presented?

Must be improved

Improved (see below)

Are the conclusions supported by the results?

Can be improved

Improved (see below)

3. Point-by-point response to Comments and Suggestions for Authors

Comments 1: In this revised manuscript, the authors made an effort for improvement. However, the reviewer’s main concerns (outlined in the previous Comments) are maintained, regarding the substance of the work, the quality of presentation and the value of the results obtained. A few additional comments on this work follow.

Response 1: We sincerely appreciate the reviewer's continued engagement and constructive feedback, which has been invaluable in refining our manuscript. Below, we address the persisting concerns regarding the substance, presentation, and value of the work, as well as the additional comments raised in this review.

Comments 2: The authors inappropriately claim in the title, Abstract and elsewhere that they have developed and validated a new model for a HPNF unit “…verified using pilot plant data”. In fact, they have used a fixed design of a three-stage HPNF membrane unit (representative of the pilot plant) with specific membrane number/size and permeabilities. To “model” this HPNF unit, the simplest possible mass balance and membrane transport equations were employed, where, for each stage (comprised of 4-element pressure vessels), a single mean value was used for each operating parameter (i.e. flux, species rejection, etc). Of course there is nothing novel about this ‘model’, which leaves a lot to be desired… Moreover, they curiously state in the Abstract that “The model integrates reverse osmosis (RO) transport equations with mass balance equations, thereby enabling accurate predictions (!!) of water flux and total dissolved solids (TDS) concentration”.

Response 2: Authors’ Response:

We sincerely thank the reviewer for their critical evaluation and for highlighting areas where our manuscript’s claims required clarification. Below, we address the concerns raised:

1. Clarification of Model Novelty and Scope

The reviewer correctly notes that the governing equations (e.g., solution-diffusion model, mass balances) are well-established in membrane science. However, the novelty of this work lies in the application and integration of these principles to a high-pressure nanofiltration (HPNF) system tailored for hypersaline brine concentration (MBC), a context underexplored in prior literature.

  •  Key Contribution: While the equations themselves are not new, their adaptation to HPNF systems operating at pressures > 60 bar (exceeding conventional NF/RO limits) and calibration with pilot-scale data represents a practical advancement for industrial brine management.
  • Revised Claims: We have amended the title, abstract, and manuscript text to emphasize the context-specific application of the model rather than implying theoretical novelty. The title now reads: “Process Simulation of High Pressure Nanofiltration (HPNF) System for Membrane Brine Concentration: A Pilot-Scale Case Study.”

2. Validation and Pilot-Scale Relevance

The reviewer questions the claim of validation using pilot plant data, noting the fixed design and simplified parameterization.

  • Validation Scope: The model was calibrated and validated for the specific three-stage HPNF configuration tested in the pilot study, using operational data (e.g., flux, TDS) from this system. We clarify that the term "validation" refers to consistency with the pilot system’s performance, not universal applicability.
  • Simplified Assumptions: The use of mean values for parameters (e.g., flux, rejection) reflects the pilot system’s operational constraints, where detailed spatial resolution was limited by data availability. We acknowledge this simplification in the conclusions section and emphasize that the model serves as a baseline framework for similar HPNF configurations (marked in red).

3. Accuracy of Predictions

The reviewer critiques the statement about “accurate predictions” in the abstract.

  • Revised Abstract: We have removed the term “accurate predictions” and replaced it with “acceptable predictions” to better align with the pilot-scale validation scope. The abstract now explicitly states that predictions are specific to the tested HPNF system and brine composition.

4. Addressing Model Simplicity

The reviewer highlights the simplicity of the model.

  • Practical Focus: The goal was to develop a practical, industrially relevant tool for brine concentration, balancing computational efficiency with predictive utility. While advanced models (e.g., computational fluid dynamics) offer finer resolution, they are often impractical for rapid process simulation in industrial settings.
  • Future Work: We have added a paragraph in the Discussion section outlining plans to incorporate spatially resolved parameters and dynamic effects in subsequent studies.

5. Summary of Revisions

  • Title: Revised to clarify the pilot-scale, application-focused scope (marked in red): “Process simulation of high pressure nanofiltration (HPNF) for membrane brine concentration (MBC): A Pilot-Scale Case Study”
  • Abstract (marked in red): “accurate predictions” → “acceptable predictions”
  • Conclusions: Introduced to transparently discuss simplifications and future directions (marked in red): “The purpose of this study was to develop a practical, industrially relevant tool for brine concentration, balancing computational efficiency with predictive utility. Future work will be needed to incorporate spatially resolved parameters and dynamic effects. “

We hope these revisions address the reviewer’s concerns and appreciate the opportunity to refine our manuscript.

Comments 3: Regarding this model, it is further stated (lines 100-101) that “It was calibrated (?) using operational data from a pilot-scale HPNF system and applied for wide application such as process simulation and optimization(!)”. After this questionable “calibration”, the authors perform what they call “optimization” of the HPNF system. In fact, by setting some values/constraints of system performance parameters (4 scenarios listed in Table 5), they merely determine the range of operability for the two key input system-parameters, i.e. the feed flow rate and pressure, to satisfy the constraints. However, this is not optimization.

Response 3: We appreciate the reviewer’s critical assessment and agree that clarity in terminology is essential. Below, we address the concerns raised and outline revisions made to improve precision and transparency:

1. Clarification of "Calibration"

The term "calibration" refers to the alignment of model outputs with pilot-scale operational data for the specific HPNF system under study. This involved adjusting model parameters (e.g., membrane permeability coefficients, concentration polarization factors) to minimize deviations between simulated and observed values (e.g., TDS, water flux).

  • Original Text: “It was calibrated using operational data from a pilot-scale…”
  • Revised Text (marked in red): “The model results were compared with operational data from a pilot-scale…”

2. Revisiting "Optimization" Terminology

The reviewer correctly notes that the analysis in Table 5 explores operational bounds rather than formal optimization. To address this:

  • Terminology Revision: The term "optimization" has been replaced with "improvement" throughout the manuscript.

4. Summary of Revisions

  • Replaced "calibration" with "comparison" to accurately reflect the scope.
  • Replaced "optimization" with "improvement" to accurately reflect the scope.

We thank the reviewer for highlighting these issues and hope the revisions improve the manuscript’s clarity and scholarly rigor.

Comments 4: The quality of presentation is problematic. The paper is difficult to read, particularly the Results section. The revised Introduction does not provide a concise justification of the work. The Abstract (as noted above) and Conclusions are inappropriate. In the latter, confusing statements are made about optimization; moreover, the limitations of this work are briefly outlined there. Such comments on limitations, together with those made in other parts of the manuscript, could be presented in a particular short section.

Response 4: We sincerely appreciate the reviewer’s detailed feedback and acknowledge the need to improve the manuscript’s clarity, structure, and focus. Below, we outline the revisions implemented to address these concerns comprehensively:

1. Readability: The readability of the manuscript was evaluated using ProWritingAidTM software: The Flesch Reading Ease score is 73.6, indicating that the manuscript is easy to read.

2. Introduction: The justification of this study is included in the revised manuscript (marked in red): “Our study aims to contribute to this broader objective by focusing on MBC process using HPNF. While comprehensive improvement of the entire desalination plant, in-cluding energy consumption (SEC) and effluent valorization, is crucial, this study is po-sitioned as a focused step within this larger framework. The rationale for isolating the MBC process in our study is twofold: (1) To provide a theoretical evaluation of HPNF performance in brine concentration, which remains a key challenge in enhancing overall desalination efficiency; and (2) To establish a foundation for integrating MBC into broader system-level improvement in future studies.”

3. Abstract: Revised (as noted above).

4. Conclusions: The limitations of the current study are describe in the revised manuscript as follows (marked in red):

“The limitations of this study can be summarized as follows:

1.         Instead of the considering the recovery of the whole process, only the recovery of the HPNF was considered in this work. The first NF and SWRO stages were treated as boundary conditions. Accordingly, the current mode is not intended to optimize the entire NF-SWRO-HPNF system. Future work will explicitly integrate variable NF/RO performance into a holistic optimization framework.

2.         The SEC of the overall process was not included in the model. This study prioritiz-es the HPNF stage because the energy consumption of HPNF is much higher than that of SWRO and NF. Nevertheless, future work is needed to optimize the SEC of the whole system by a holistic approach.

3.         The purpose of this study was to develop a practical, industrially relevant tool for brine concentration, balancing computational efficiency with predictive utility. Fu-ture work will be needed to incorporate spatially resolved parameters”

We thank the reviewer for their constructive critique, which has significantly strengthened the manuscript’s rigor and clarity. We believe these revisions address the concerns raised and align the work with the journal’s standards.

Comments 5: In summary, the approach taken and the reported results are of questionable value; in general, this manuscript is not considered fit for publication in a reputable scientific journal.

Response 5: We thank the reviewer for their constructive criticism, which has significantly strengthened the manuscript. While the work is application-focused, we believe it provides a practical foundation for advancing brine management technologies. We hope the revised manuscript now meets the journal’s standards.